# Legacies of temperature fluctuations promote stability in marine biofilm communities

Luca Rindi [1] ✉, Jianyu He[1,3], Mara Miculan [2,4], Matteo Dell'Acqua[2], Mario Enrico Pè[2] & Lisandro Benedetti-Cecchi [1]

The increasing frequency and intensity of extreme climate events are driving significant biodiversity shifts across ecosystems. Yet, the extent to which these climate legacies will shape the response of ecosystems to future perturbations remains poorly understood. Here, we tracked taxon and trait dynamics of rocky intertidal biofilm communities under contrasting regimes of warming (fixed vs. fluctuating) and assessed how they influenced stability dimensions in response to temperature extremes. Fixed warming enhanced the resistance of biofilm by promoting the functional redundancy of stress-tolerance traits. In contrast, fluctuating warming boosted recovery rate through the selection of fast-growing taxa at the expense of functional redundancy. This selection intensified a trade-off between stress tolerance and growth further limiting the ability of biofilm to cope with temperature extremes. Anticipating the challenges posed by future extreme events, our findings offer a forward-looking perspective on the stability of microbial communities in the face of ongoing climatic change.

Ongoing changes in the Earth's climate entail more frequent and intense extreme events which are already modifying the structure and function of many ecosystems[1,2]. However, most studies on extreme events have focused on assessing the impact of current climate modifications, whereas less attention has been given to how exposure to past climate anomalies shapes the responses of ecological communities to future perturbations[3,4]. Extremes in weather patterns and climate variability are predicted to alter the taxonomic and functional diversity of ecosystems, thereby impairing their ability to deal with future perturbations[5,6]. Identifying the properties that underpin the stability of ecological systems under future climate change scenarios is a central focus of contemporary research in many fields[7].

Past climatic fluctuations can fundamentally shape various dimensions of stability of ecological communities – resistance, resilience, and temporal stability–by driving changes in species and functional composition[8]. Exposure to environmental variability can foster species coexistence through temporal niche differentiation, enhancing functional and taxonomic diversity[9,10]. Both theoretical and empirical evidence indicate that communities with functionally redundant species, each with different sensitivities to environmental conditions, are better equipped to buffer perturbations ("insurance effect")[11,12]. In species-rich communities, recovery may be facilitated by fast-growing taxa or enhanced through the establishment of positive interactions[13,14]. Yet, by promoting asynchronous fluctuations–where the decline in some species is offset by the increase in others–functionally redundant communities can stabilize community-level properties (e.g., biomass) over time[12]. Conversely, exposure to intense environmental fluctuations may favor stress-tolerant taxa, filtering out the more sensitive ones. This selection can enhance community resistance at the cost of resilience due to trade-offs between stress

[1]Department of Biology, University of Pisa, Pisa, Italy. [2]Institute of Life Sciences, Scuola Superiore Sant'Anna, Pisa, Italia. [3]Present address: Marine Science and Technology College, Zhejiang Ocean University, Zhoushan City, Zhejiang, China. [4]Present address: Center of Excellence for Sustainable Food Security, Biological and Environmental Sciences and Engineering Division (BESE), King Abdullah University of Science and Technology (KAUST), Thuwal, Saudi Arabia. ✉e-mail: luca.rindi@unipi.it

tolerance and growth rates[10,15]. Communities dominated by stress-tolerant, slow-growing taxa may maintain greater stability over time and effectively buffer extreme events, but they may exhibit slower recovery following perturbations[16,17]. Despite considerable progress in both theoretical and empirical research, our understanding of how historical legacies shape the response of ecological systems to future climate-change scenarios remains limited.

Here, we performed a field experiment to assess how exposure to relatively stable *vs.* fluctuating warming conditions shaped the taxonomic and functional diversity of rocky shore biofilm, a community of cyanobacteria and microalgae living embedded in a matrix of self-produced polymeric substances[18]. We further investigated how changes in the biofilm community driven by previous exposure to warming affected multiple stability dimensions in response to subsequent temperature extremes (Supplementary Fig. 1a). Three reasons make rocky intertidal biofilm an ideal system for field experiments: 1) the short-generation times of constituent taxa allow for multigeneration field-experiments[18], 2) the assessment of the aggregated status of the community through nondestructively remote-sensing techniques (e.g., multispectral NIR-cameras)[19,20] and 3) its suitability to genomic analysis that readily informs on the structure and functioning of the community[21]. Biofilm also plays a key role in coastal habitats, contributing to primary productivity, providing food for primary consumers, and influencing the settlement of many invertebrates[18].

We expected that exposure to a thermally fluctuating environment would maintain higher taxonomic diversity and functional redundancy than a more thermally uniform one. Greater functional redundancy is expected to enhance the community's ability to buffer perturbations and maintain temporal stability in aggregate biomass and ultimately provide a rapid recovery driven by fast-growing taxa (Fig. 1a, b). However, if thermal fluctuations exceed the physiological thresholds of sensitive taxa, environmental filtering may favor the selection of thermally resistant ones. This directional selection could reduce both diversity and functional redundancy while enhancing the biofilm's ability to withstand climate extremes[22] (Fig. 1b). Nonetheless, given the trade-off between stress tolerance and growth ability[23], increased resistance might come at the cost of a diminished capacity to recover from temperature extremes.

## Results

A two-phase field experiment was performed to assess how warming history shaped the structure of the biofilm community and its stability dimensions in response to subsequent perturbations (Fig. 1a, b). In the first phase, we imposed contrasting temporal regimes of warming (fluctuating-warming pulses *vs.* fixed-warming pulses) to the biofilm community. In the second phase, we evaluated how the warming history affected biofilm stability in response to two consecutive extreme temperature events (60 °C), simulating exceptionally rare thermal events − with a probability of occurrence less than 1% (Fig. 1a–c). To achieve this, we exposed plots naturally colonized by biofilm to four experimental treatments over a period of four months: 1) control, with biofilm exposed to natural temperature variability; 2) fixed, repeated pulses of 12 °C above the ambient temperature (SD = 0) (Fig. 1a, c, d and Supplementary Fig. 1b); 3) fluctuating, in which temperature varied at each pulse application, producing multiple fluctuating treatments (fluct-s1, -s2 and -s3) with the same mean temperature as in the fixed-warming condition (12 °C above the ambient temperature), but with a different variance (SD = 5 °C; Fig. 1d) (see Methods) and 4) extreme-only treatment consisting of plots that only received the extreme temperature events. This latter treatment was designed to isolate the effects of warming history from those of extreme temperatures. Temperatures used in fixed and fluctuating treatments represent moderate thermal events, occurring with a higher likelihood than extreme temperatures (probabilities ranging from 40% to 4%) (Fig. 1c). Notably, as it was a field experiment, natural variation in

temperature conditions influenced all plots, regardless of their assigned treatment. Thus, the fixed-warming treatment superimposed a temporally consistent pattern of heating events over naturally fluctuating conditions, but without eliminating natural variations (Fig. 1c, d).

## Impact of fluctuating warming on biofilm diversity

The whole-genome sequencing and assembly identified 137 Metagenome Assembled Genomes (MAGs) with a high genome completeness (mean = 88.01%, 95%CIs = 72.14−99.22%) and low levels of contamination (mean = 2.69%, 95%CIs = 0.53−4.77%), mostly of which belonged to the *Protobacteria*, *Bacteridota*, *Acidobacteria*, and *Cyanobacteria* phyla (Supplementary Fig. 2). Contrary to our hypothesis, we found that exposure to a fixed regime of warming enhanced Shannon diversity (measured as Hill number) compared to the of fluctuating regime ($z = 2.42$, $p < 0.05$, Fig. 2b, Supplementary Table 1), with this effect becoming more pronounced over time ($t = 2.18$, $p < 0.05$, Fig. 2b, Supplementary Table 1). Simpson diversity (q = 2) consistently paralleled Shannon diversity, exhibiting a diverging trend that culminated in a significant deviation from controls ($t = -2.93$, $p < 0.05$) and the fluctuating treatment ($t = -2.50$, $p < 0.05$) (Fig. 2c, Supplementary Fig. 3 and Supplementary Table 1). There were no significant differences in richness (q = 0) across warming regimes over the experiment, except at the end of the study, when richness tended to be lower in the fluctuating than in the fixed-warming treatment ($t = 1.81$, $0.08 > p > 0.05$, Fig. 2a, Supplementary Table 1). Richness, Shannon, and Simpson diversity did not differ between controls plots and those that received only temperatures extremes (Fig. 2a–c).

We conducted an additional analysis using assembled 16S rRNA contigs to better capture rare taxa in the biofilm community. This analysis produced results that were qualitatively consistent with those from MAGs across all diversity metrics (Supplementary Fig. 4). Notably, it revealed that fluctuating warming led to a significantly divergent pattern from fixed warming in richness, Shannon and Simpson diversity over the course of the study (Supplementary Table 2). Furthermore, no differences were observed between controls and extreme-only plots across all diversity metrics before the application of extreme events (Supplementary Fig. 5, Supplementary Table 3), nor between controls and artifact controls − unwarmed but shaded plots (see Methods) (Supplementary Table 4).

To further test the hypothesis that fluctuating warming should drive asynchronous fluctuations among taxa, we compared MAG synchrony and average MAG-level population variability among warming conditions (Fig. 1a). Unexpectedly, we found that warming increased synchronous temporal fluctuations among MAGs regardless of temporal regime (Cont. *vs.* Warmed; $z = 2.56$, $p < 0.05$), but reduced average population variability (Cont. *vs.* Warmed; $z = -4.31$, $p < 0.001$) compared to controls, which were characterized by asynchronous fluctuations and low average population variability (Fig. 2c, d, Supplementary Fig. 6 and Supplementary Table 5).

Next, we assessed the relative contributions of stochastic (e.g., random birth-death events) *vs.* deterministic processes (e.g., environmental filtering) in shaping the composition of biofilm community using the Normalized Stochasticity Ratio (NST)[24]. First, this analysis revealed that under natural conditions (controls) the relative contribution of stochastic processes increased over time (Fig. 3). In contrast, stochastic processes appeared less important, compared to controls, when biofilm was exposed to both fixed- and fluctuating-warming regimes (31% and 36%, respectively; Con. *vs.* Fixed and Con. *vs.* Var. $p < 0.05$, Fig. 3, Supplementary Table 6). Furthermore, in the first phase of the experiment extreme-only plots did not show a significant deviation from controls ($p > 0.25$, Supplementary Fig. 7).

Overall, warming − regardless of its temporal patterns − acted as a deterministic filter on biofilm communities, reducing its stochasticity. Fixed warming pulses increased biofilm diversity compared to

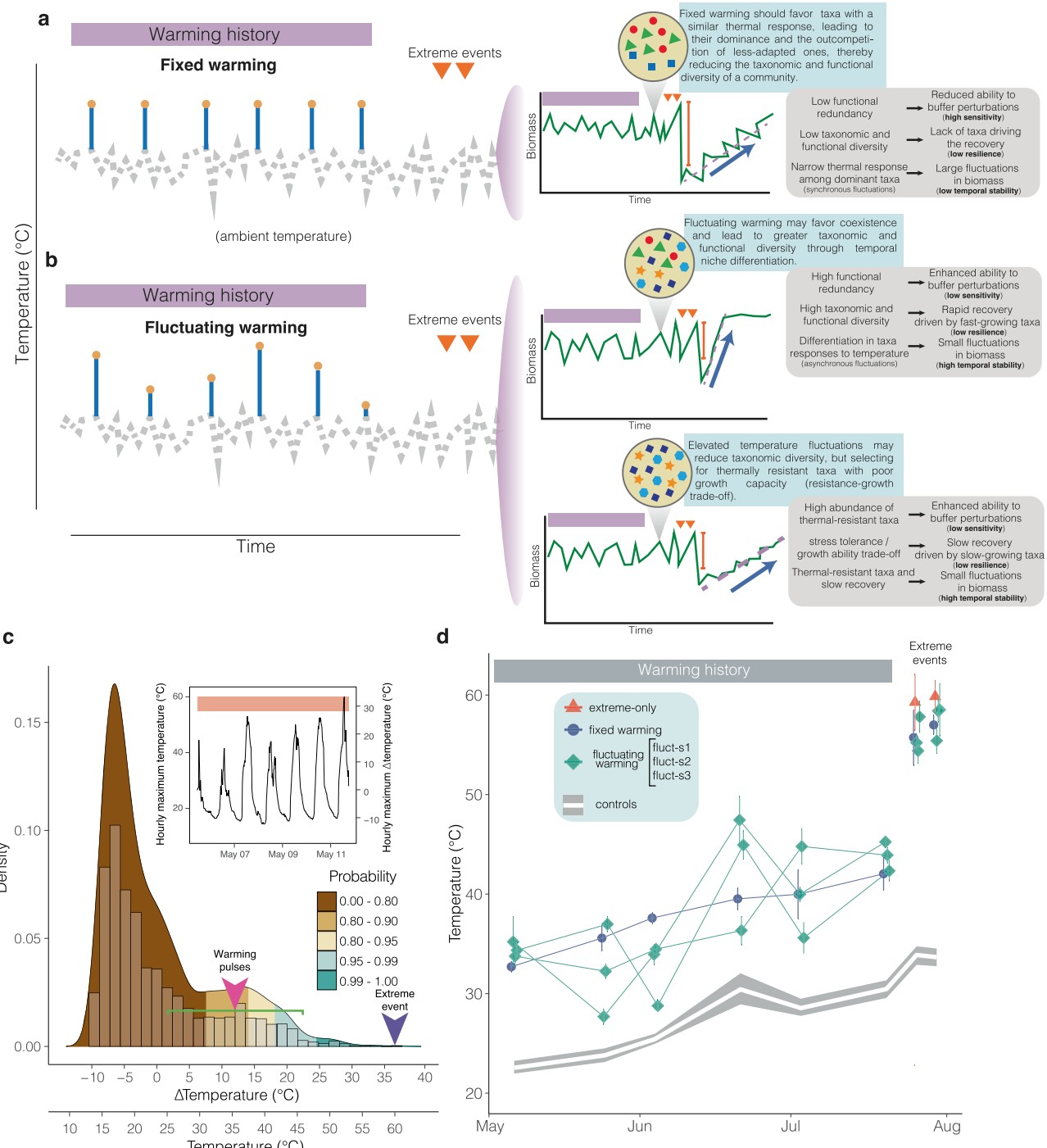

**Fig. 1 | Thermally fluctuating environments may stabilize biofilm communities against perturbations by promoting coexistence and functional redundancy or through the selection of resistant taxa.** Conceptual diagram illustrating how different warming regimes shape biofilm community diversity − taxonomic and functional − and different stability dimensions. **a** Under steady warming, competitive hierarchies emerge, allowing narrowly adapted taxa to dominate and outcompete less specialized ones. This reduces taxonomic and functional diversity, resulting in a less diverse community that is highly sensitive to perturbations, poorly resilient, and less temporally stable. **b** Fluctuating warming may exert two opposing forces on a community. On the one hand, it may promote coexistence through temporal niche differentiation that promotes asynchronous fluctuations among taxa, thereby relaxing interspecific competition. This should lead to a more diverse, functionally redundant community that is temporally stable, resistant to perturbations (low sensitivity), and highly resilient (quick recovery). On the other hand, large temperature fluctuations may filter out thermally susceptible species. This should reduce interspecific competition and favors thermally resistant taxa,

resulting in a less diverse community that is resistant to perturbations, stable over time, but less resilient due to a trade-off between growth and resistance. The grey-dashed line indicates ambient temperature, orange circles represent temperature pulses, and orange down-facing triangles indicate extreme events. **c** Density plot and histogram showing the distribution of hourly maximum air temperatures collected at the study site during May and June of 2015 and 2017. The black line represents the fitted kernel density estimate, while the colored areas mark the quantiles. The purple down-facing arrow marks the average temperature warming pulses, while the green line encompasses the range of temperatures used in the fluctuating treatments. The inset depicts the time series of hourly maximum air temperatures, with temperatures reaching 60 °C during the central hours of the day. **d** Time series of air temperatures −collected with temperature loggers − under different warming regimes (mean ± SE; $n = 4$). Ambient temperature (outside the warming chamber) is shown as a 95% confidence interval in light grey (mean ± 95% CI, $n = 16$). Source data are provided as a Source Data file.

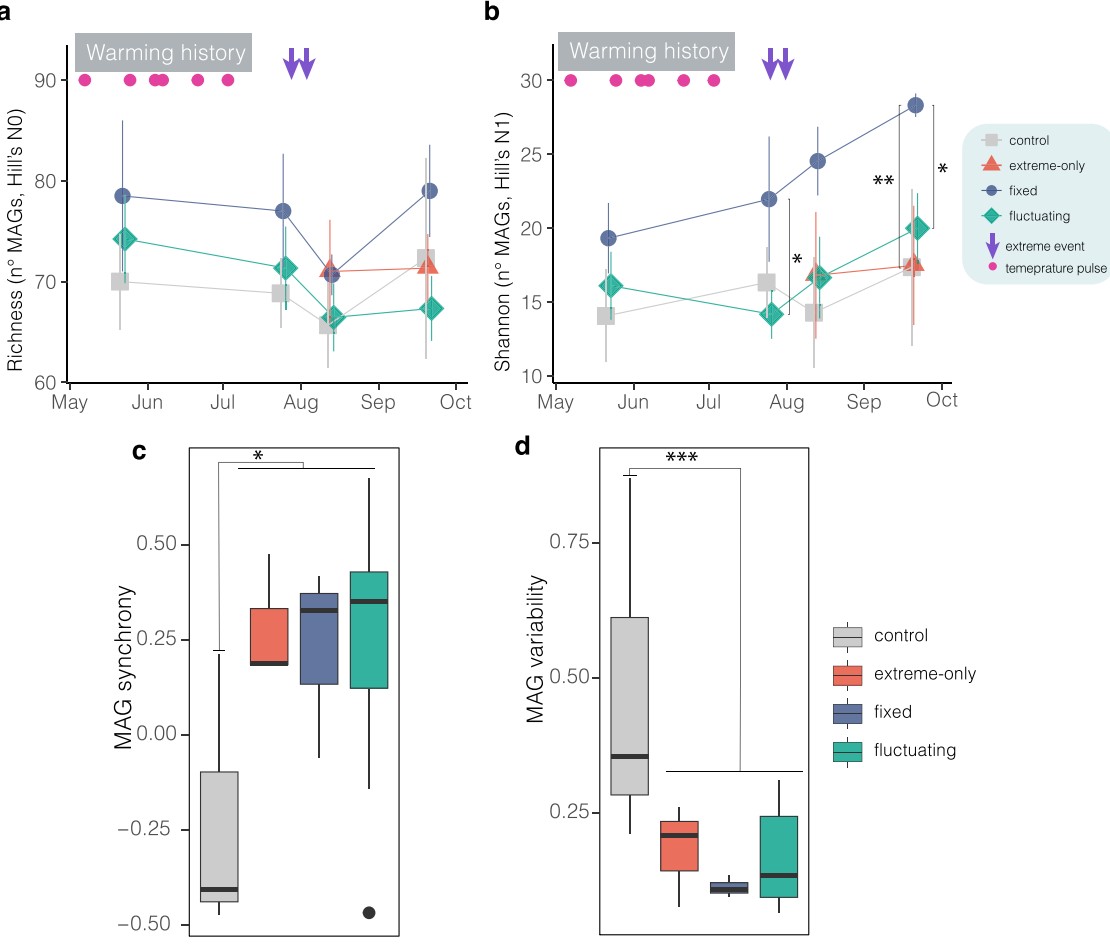

**Fig. 2 | Fixed warming increased the diversity of biofilm community.** Time-series of (**a**), richness (# MAGs) and (**b**), Shannon diversity (Hill number of order 1). Boxplots of (**c**), weighted asynchrony and (**d**), weighted average variability of the populations of the community's constituent MAGs. For (**a**, **b**), error bars indicate the standard errors of the mean (controls and extreme-only treatments, $n = 3$; fixed-warming treatment, $n = 2$ at the first sampling and $n = 3$ thereafter). Prior to the imposition of extreme temperatures, the control condition represents the average of six plots ($n = 6$; three extreme-only and three control plots), while the fluctuating treatment is shown as the average of nine plots ($n = 9$) derived from three fluctuating sequences (fluct-s1, fluct-s2, and fluct-s3). Separate trajectories for the control and extreme-only treatments are provided in Supplementary Fig. 5. In (**c**, **d**), box-plots display the median (central line), interquartile range (25th–75th percentiles), whiskers (5th and 95th percentiles), and outliers (individual points; $n = 3$ or 9). Statistical significance is indicated as *$p < 0.05$, **$p < 0.01$, and ***$p < 0.001$; Wald $z$-tests were used for fixed effects in (**a**, **b**) (with $p$-values adjusted for multiple comparisons at the final sampling date using the false discovery rate method [FDR]), and one-way ANOVA $F$-tests for (**c**, **d**). Pink circles denote temperature pulses, while purple down-facing arrows indicate extreme temperature events. All statistical values are provided in Supplementary Table 1 (available in the Source Data). Source data are provided as a Source Data file.

controls, whereas fluctuating warming reduced richness and Shannon diversity, particularly at the expense of rare taxa.

## Potential functions and growth-rate of biofilm communities

To determine how warming history shaped the functional structure of the dominant members of biofilm community, we identified potential stress-tolerance traits by mapping genome sequences into a hierarchical trait space. We then assessed functional group richness and functional group redundancy (functional over-redundancy; FOR) of stress-tolerance-related traits[25]. Although fixed and fluctuating warming had no effect on functional group richness (Fig. 4a, Supplementary Table 7), exposure to fixed warming increased functional redundancy relative to control ($z = 3.60$, $p < 0.001$) and to fluctuating warming $z = 3.08$, $p < 0.01$, Fig. 4b, Supplementary Table 7). In the fixed-warming treatment, FOR converged toward control values following extreme warming ($t = -0.67$, $p > 0.54$), whereas fluctuating warming tended to induce a diverging trend from the controls and the fixed-warming treatment. Finally, there was no significant difference in FOR between the control and extreme-only conditions (Fig. 4b, Supplementary Table 8).

We also calculated functional vulnerability (FVULN), which is the proportion of functional groups in an assemblage represented by a single MAG[25]. FVULN showed an opposite temporal pattern compared to FOR: exposure to fixed warming reduced FLVUN compared to controls ($z = -3.97$, $p < 0.001$) and to the fluctuating treatment ($z = -4.18$, $p < 0.001$, Fig. 4c, Supplementary Table 7). In contrast, FVULN maintained elevated values throughout the experiment under fluctuating temperatures (Fig. 4c). These findings suggested that, alongside promoting taxonomic diversity, fixed warming promoted functional redundancy of stress-tolerance-related traits. In contrast, fluctuating warming tended, in the long run, to reduce functional redundancy and to enhance the functional vulnerability of the biofilm community.

To assess how warming regimes influenced the biofilm's resilience to extreme temperatures, we used genomic data to estimate the minimal doubling time of each MAG[26,27]. We then calculated the Community Weighted Minimal Doubling Time (CWMDT) to quantify the distribution of fast- and slow-growing taxa within the community. We found that CWMDT diverged progressively from controls under fluctuating warming (temporal trend: $z = 2.32$, $p < 0.05$; last sampling

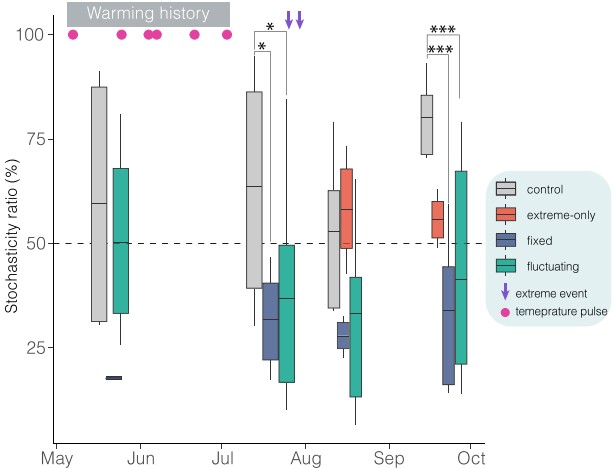

**Fig. 3 | Fixed and fluctuating warming reduced the overall-community stochasticity.** Time series of the phylogenetic Normalized Stochasticity Ratio (NST) under different temporal warming regimes. In the boxplots, the central line indicates the mean, the box spans the interquartile range (25th–75th percentiles), whiskers extend to the 10th–90th percentiles and individual points represent outliers. For controls and extreme-only treatments, $n = 3$; for the fixed warming treatment, $n = 2$ at the first sampling and $n = 3$ thereafter. Before the imposition of extreme temperatures, the control condition is shown as the average of six plots ($n = 6$; three extreme-only plus three control plots), while the fluctuating treatment is the average of nine plots ($n = 9$) from three fluctuating sequences (fluct-s1, fluct-s2, and fluct-s3). Separate trajectories for the control and extreme-only treatments are shown in Supplementary Fig. 7. Values in the boxplots are derived from n(n–1)/2 pairwise NST comparisons (see Methods). Statistical significance is indicated by asterisks (*$p < 0.05$, **$p < 0.01$, ***$p < 0.001$) based on a one-sided bootstrap test ($n = 1000$), with $p$-values adjusted for multiple comparisons at each sampling date using the false discovery rate (FDR) procedure. To enhance clarity, significant comparisons are reported only for the 2nd and 4th sampling dates. All statistical values appear in Supplementary Table 6 (available in the Source Data), and source data are provided as a Source Data file. Pink circles denote temperature pulses, and purple down-facing arrows mark extreme temperature events.

date: $t = 3.62$, $p < 0.01$), indicating a marked shift towards fast-growing taxa in the fluctuating treatment (Fig. 4d, Supplementary Table 7). In addition, CWMDT declined significantly in the fixed-warming treatment in response to extremes, diverging from the control ($t = 2.42$, $p < 0.05$) and converging towards the extreme-only treatment (Fig. 4d). Finally, no differences were detected across all functional metrics and minimal doubling time between controls and extreme-only plots before the imposition of extreme events (Supplementary Fig. 8, Supplementary Table 8), nor between controls and artifact controls (Supplementary Table 9).

We assessed the trade-off between stress tolerance and growth ability in MAGs by examining the relationship between stress-tolerance gene modules (used as a proxy for stress tolerance) and minimal doubling time (MDT). A significant positive relationship between the number of stress tolerance-related traits and MDT was observed across all MAGs (Fig. 5a) ($z = 2.62$, $p < 0.01$, Supplementary Table 10). To assess the impact of warming history on the relationship between MDT and stress tolerance, we examined whether this relationship persisted in MAGs that were significantly more abundant under fluctuating and fixed warming conditions compared to controls (Supplementary Fig. 9). The relationship persisted under fluctuating warming (Fig. 5b) ($z = 4.18$, $p < 0.001$) but vanished under fixed warming (Fig. 5c) ($z = 1.51$, $p > 0.11$) (Supplementary Table 10). These findings indicate that fluctuating and extreme temperatures favored fast-growing taxa over slow-growing ones, which were nonetheless constrained by the trade-off between stress tolerance and growth ability.

## Multiple dimensions of stability in response to extreme temperatures

We assessed how taxonomic and functional shifts in the biofilm community affected its ability to cope with extreme warming events by measuring four dimensions of stability: sensitivity (inverse of resistance), resilience, temporal stability, and recovery (Fig. 6a). Stability components were derived using biofilm biomass acquired after exposure to extremes events and setting controls as the baseline[28].

The analysis of stability components revealed that prior exposure to both fixed and fluctuating warming had a stabilizing effect on the biofilm community, diminishing its sensitivity to subsequent warming extremes (Fig. 6b, c, Supplementary Table 11). In the fixed-warming scenario, this diminished sensitivity stemmed from biomass accumulation occurring during the first phase of the experiment ($z = 2.18$, $p < 0.05$, Supplementary Table 12), which later declined to control levels in response to extreme temperatures (Fig. 6b, c, Supplementary Table 11). In contrast, biofilm exposed to fluctuating warming consistently mirrored controls throughout the initial phase of the study, exhibiting no significant alterations in response to subsequent warming extremes (Fig. 6b, c, Supplementary Tables 11, 12). Crucially, plots without a warming history (extreme-only) initially exhibited biomass levels comparable to controls (Fig. 6b, Supplementary Table 12); however, once exposed to extreme temperatures they showed a significant deviation from controls (Sensitivity: $z = -3.12$, $p < 0.01$, to Fig. 5b, c, Supplementary Table 11). Despite this initial deviation, the extreme-only treatment swiftly rebounded toward control levels (Resilience: $z = 2.40$, $p < 0.05$; Fig. 6b, e, Supplementary Table 16). Post-extreme events, biofilm subjected to fluctuating warming exhibited an upward trend (Resilience: $t = 2.85$, $p < 0.01$, Supplementary Table 11), surpassing biomass values observed in the controls (Fig. 6b, c). In contrast, biofilm under exposed to fixed warming did not exhibit a significant departure from controls after the imposition of extreme events ($z = 1.19$, $p > 0.20$). Finally, the 95% confidence intervals of temporal stability estimated from the controls encompassed the mean values of all the treatments, indicating the absence of any effects of warming on temporal stability (Fig. 6d). Artifacts due to the heating chambers were not detected (Supplementary Table 13).

The reduced sensitivity of warmed plots to extreme temperatures could be due to thermal acclimatization. To investigate acclimatization's role in extreme event sensitivity, we derived the relationship between the log-response ratio before and after the temperature extremes and the magnitude of last temperature pulse. However, the absence of a discernible relationship suggests that acclimatization may not be the key factor in biofilm responses to extreme temperatures (Supplementary Fig. 10).

## Discussion

Our findings provide experimental evidence of how warming legacies shape taxonomic and functional diversity of the dominant members of microbial communities and how such changes determine stability dimensions in response to temperature extremes. We found that warming history contributed to biofilm stability in two distinct ways, contingent upon the degree of temperature fluctuations experienced. Contrary to our initial hypothesis, our findings revealed that fixed warming played a crucial role in augmenting biomass, thereby enhancing biofilm's capacity to withstand extreme temperatures. Although applied at a constant rate, multiple pulses of the same magnitude produced slightly greater variability than the ambient ones (controls) (Fig. 1d). In line with prior research[10,29,30], this slight increase in variability not only promoted coexistence but also enhanced primary production within the biofilm community, boosting both diversity and the functional redundancy of stress-tolerance traits in its dominant members. Highly fluctuating warming acted as a filter on the biofilm community by reducing its diversity and promoting fast-growing taxa (shorter generation time) over slow-growing ones

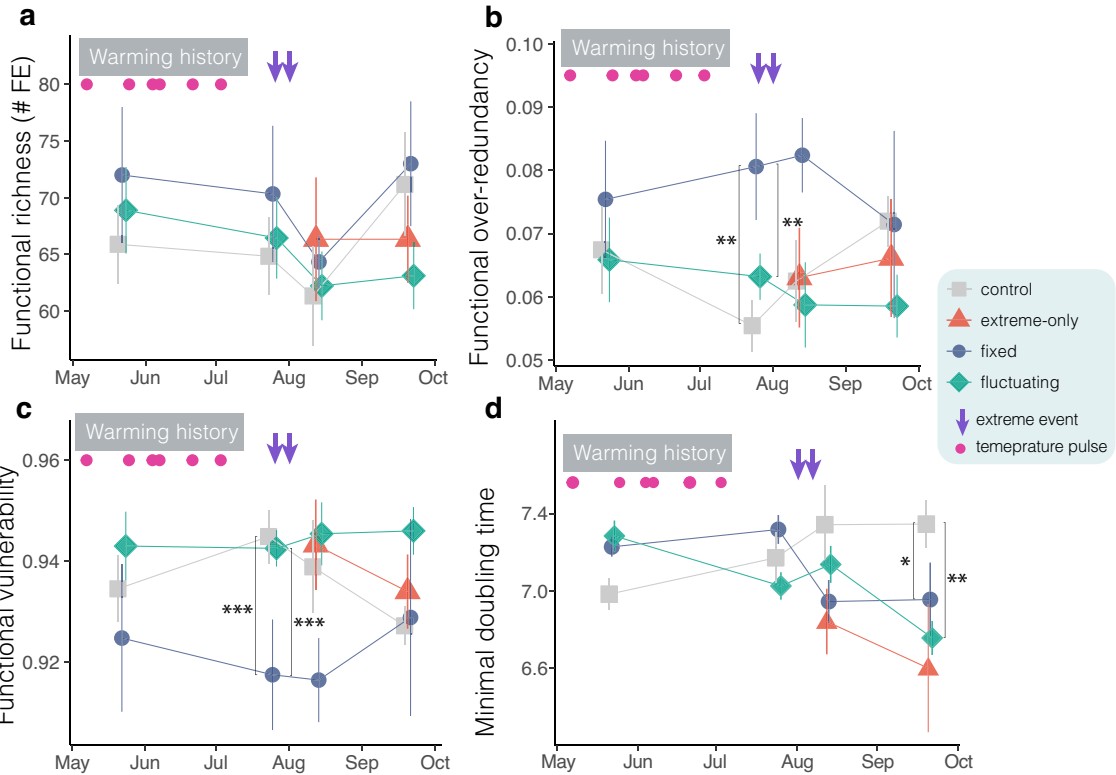

**Fig. 4 | Fixed warming enhanced the potential functional redundancy of stress-tolerance traits, while fluctuating warming promoted faster-growing taxa.**
**a** Time series of functional group richness (FR; number of functional entities [FE]), **b** functional over-redundancy (FOR), **c** functional vulnerability (FVuln) of stress-tolerance traits of biofilm community, and **d** Minimal Doubling Time (MDT). Error bars indicate the standard errors of the mean (controls and extreme-only treatments, $n = 3$; fixed warming treatment, $n = 2$ at the first sampling and $n = 3$ thereafter). Prior to the imposition of extreme temperatures, the control condition represents the average of six plots ($n = 6$; three extreme-only and three control plots), while the fluctuating treatment is shown as the average of nine plots ($n = 9$) derived from three fluctuating sequences (fluct-s1, fluct-s2, and fluct-s3). Separate trajectories for the control and extreme-only treatments are provided in Supplementary Fig. 8. $p < 0.05$, **$p < 0.01$, ***$p < 0.001$; Wald $z$-tests for fixed effects with $p$-values adjusted for multiple comparisons at the last sampling date using the false discovery rate (FDR) method. Pink points indicate the temperature pulses, while purple down-facing arrows the extreme temperature events. All statistical values are provided in Supplementary Table 7, available in the Source Data. Source data are provided as a Source Data file.

(Figs. 4, 6d). This selection intensified the trade-off between stress tolerance and growth, likely constraining the ability of biofilms to cope with climate extremes. As a result, the reduced sensitivity to extreme events and the divergence from controls may stem from the rapid recovery observed between the onset of extreme temperatures and the first post-extremes sampling (Fig. 5a). These findings underscore the critical importance of historical conditions in shaping the responses of ecological communities to future climatic perturbations.

The reduced sensitivity to extreme temperatures observed in response to warming may stem from both thermal adaptation and acclimatization[31]. Given that thermal adaptation in microbial communities typically unfolds over several months[32,33], our 2.7-month warming exposure likely precluded significant mutation and recombination, thus ruling out rapid evolution. Furthermore, thermal adaptation in microbial communities has been usually observed in response to long-term changes in environmental factors (e.g., temperature) rather than repeated pulses, as in our experiment[34]. In contrast to adaptation, thermal acclimatization in microbial communities involves the activation or upregulation of specific gene pathways, mechanisms that generally operate on timescales of minutes to days[35]. Since a one-week interval separated warming exposure from the application of extreme events, any acclimatization effects likely dissipated during this period. In addition, the lack of correlation between biomass changes that occurred during the interval between the end of warming history and the temperature extremes, further indicates that our findings are primarily driven by a shift in biofilm diversity rather than by

acclimatization (Supplementary Fig. 10). Our findings align with recent studies showing that temperature-induced shifts in the taxonomic and functional diversity of microbial communities are key drivers of community-level responses to climate fluctuations[36,37].

We found that exposure to fluctuating warming conditions and extreme temperatures favored fast-growing taxa over slow-growing ones. A plausible explanation is that, under fluctuating warming, fast-growing taxa may be favored because of their ability to track unpredictable fluctuations more effectively than slow-growing taxa[38]. In unpredictable environments, this ability may have enhanced the fitness of fast-growing taxa, leading them to outcompete the slower-growing ones. This result concurs with theoretical and empirical studies showing that environmental fluctuations favored rare species characterized by fast growth rates[10,39]. Yet, our finding appears to contrast with a recent observational study showing that warming favored slow-growing taxa in marine microbial communities[40]. One essential difference between the studies is that the latter has been conducted at naturally fluctuating temperatures, including both warming and cooling, whereas our treatments operated only in one direction by elevating temperature above ambient levels. Cold spells could therefore favor slow-growing taxa ultimately impairing the recovery of microbial communities to perturbations. A better understanding of how temperature variability influences the distribution of fast- and slow-growing taxa in communities is, therefore, necessary to predict ecosystem responses under climate change scenarios falling outside historical ranges.

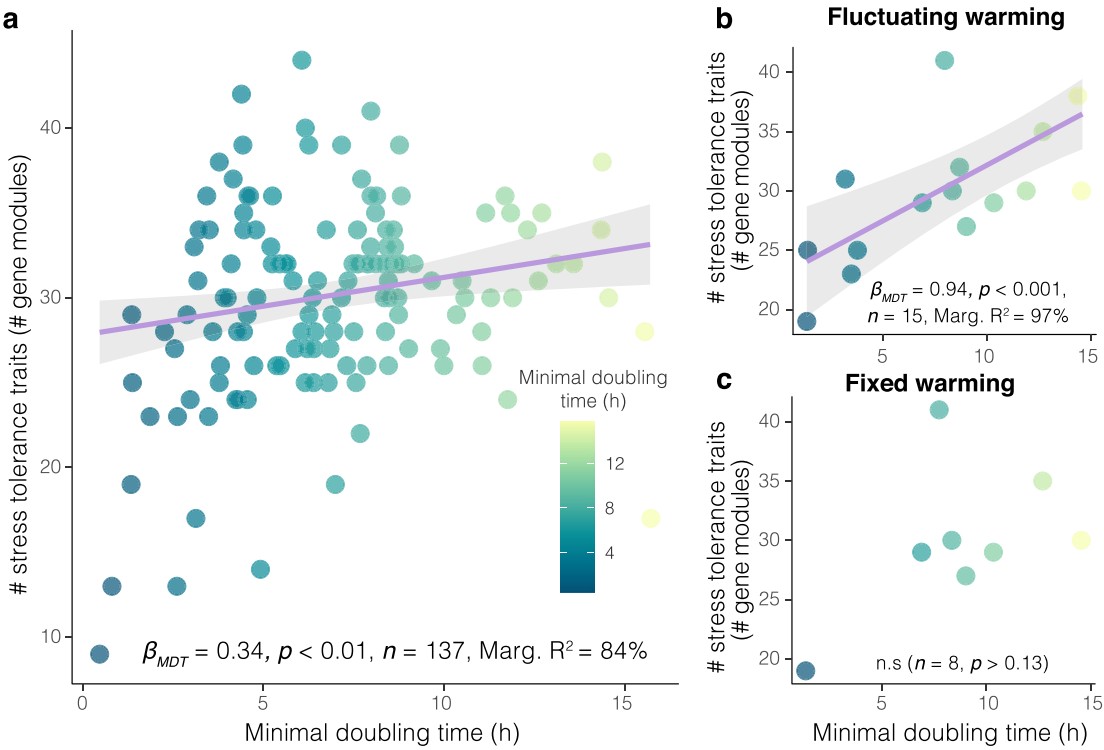

**Fig. 5 | Fluctuating warming favors taxa characterized by a positive relationship between stress tolerance traits and minimal doubling time. a** The relationship between stress tolerance traits (number of gene modules) and minimal doubling time (MDT) across all MAGs ($n = 137$). **b**, **c** the relationship for MAGs that were more abundant in fluctuating ($n = 15$) and fixed ($n = 8$) warming scenarios compared with control, respectively, as determined by differential abundance analysis (Supplementary Fig. 9). The purple line represents the linear trend, and the shaded area shows the 95% confidence intervals (95% CIs). All statistical values are provided in Supplementary Tables 10, available in the Source Data. Source data are provided as a Source Data file.

Trade-offs between stress tolerance and recovery ability are well-documented in microbial communities and arise from inherent metabolic constraints[41]. The energy demands required to maintain resistance (e.g., biofilm formation) can limit the capacity for rapid recovery (e.g, prioritization of biosynthesis), as resources needed for swift reproduction and growth following disturbances become constrained[42]. Fluctuating thermal conditions tended to intensify this trade-off within dominant taxa, indicating an adaptive advantage for marine biofilms communities that prioritize a single strategy. In the context of near-future climate change, high resilience will be crucial for ensuring rapid recovery from disturbances. However, because ecological communities face multiple stressors, gaining resistance to one stressor may come at the expense of resistance to another[43]. Future research should focus on disentangling these trade-offs among tolerances to multiple stressors and understanding their contributions to various components of stability.

The opposing contributions of asynchrony and average population variability underlie the lack of an effect of warming regimes on temporal stability (Fig. 6e). Previous studies have shown that these two mechanisms can balance each other, dampening temporal variation in aggregated variables (e.g., biomass)[44]. We indeed found that high species variability coincided with asynchronous fluctuations among MAGs, whereas synchronous fluctuations corresponded to low species variability (Supplementary Fig. 6). Species synchrony and population variability may therefore act in concert or in opposition to dampen or amplify the community-level variability of ecological communities[45]. These findings collectively highlight that warming can significantly alter stability mechanisms while maintaining the overall stability of ecological communities.

As climate fluctuations intensify and extreme events become more frequent, the stability of ecological systems in response to future extreme events is likely to be contingent on the recent history of climate anomalies[28,46,47]. Based on our study, relatively constant warming – contrary to the prevailing expectation (Fig. 1a) – fostered resistance through diversity and functional redundancy in dominant taxa. This outcome indicates that even a slight degree of warming variability may create alternating conditions that promote taxa coexistence. In contrast, fluctuating warming appears to increase biofilm sensitivity to further warming by favouring the dominance of fast-growing taxa. In a warmer and more unpredictable world, marine biofilms would be, on one side, more resilient against extreme temperatures but, on the other, poorly equipped to resist environmental perturbations and less capable of withstanding temperature extremes due to the inherent trade-off between stress tolerance and growth rate. Prolonged and sustained changes in the structure of marine biofilms are likely to lead to cascading effects on the integrity of food webs and essential ecosystem functions (e.g., nutrient cycling)[48]. Addressing the impending challenges of future extreme events, our study delivers a novel perspective, highlighting the stability and adaptability of marine biofilm communities in an era of ongoing climate change. However, further research is needed to determine whether the stability-enhancing effects of fluctuating warming extend across diverse microbial systems and microbiomes.

## Methods
### Study site
The study was performed along the coast of Calafuria (Livorno, 43° 30'N, 10°19' E) between May and September 2018. The coast is composed of gently sloping sandstone platforms with high-shore levels (0.3–0.5 m above mean low water level) characterized by assemblages of barnacles interspersed among areas of seemingly bare rock, where the biofilm develops.

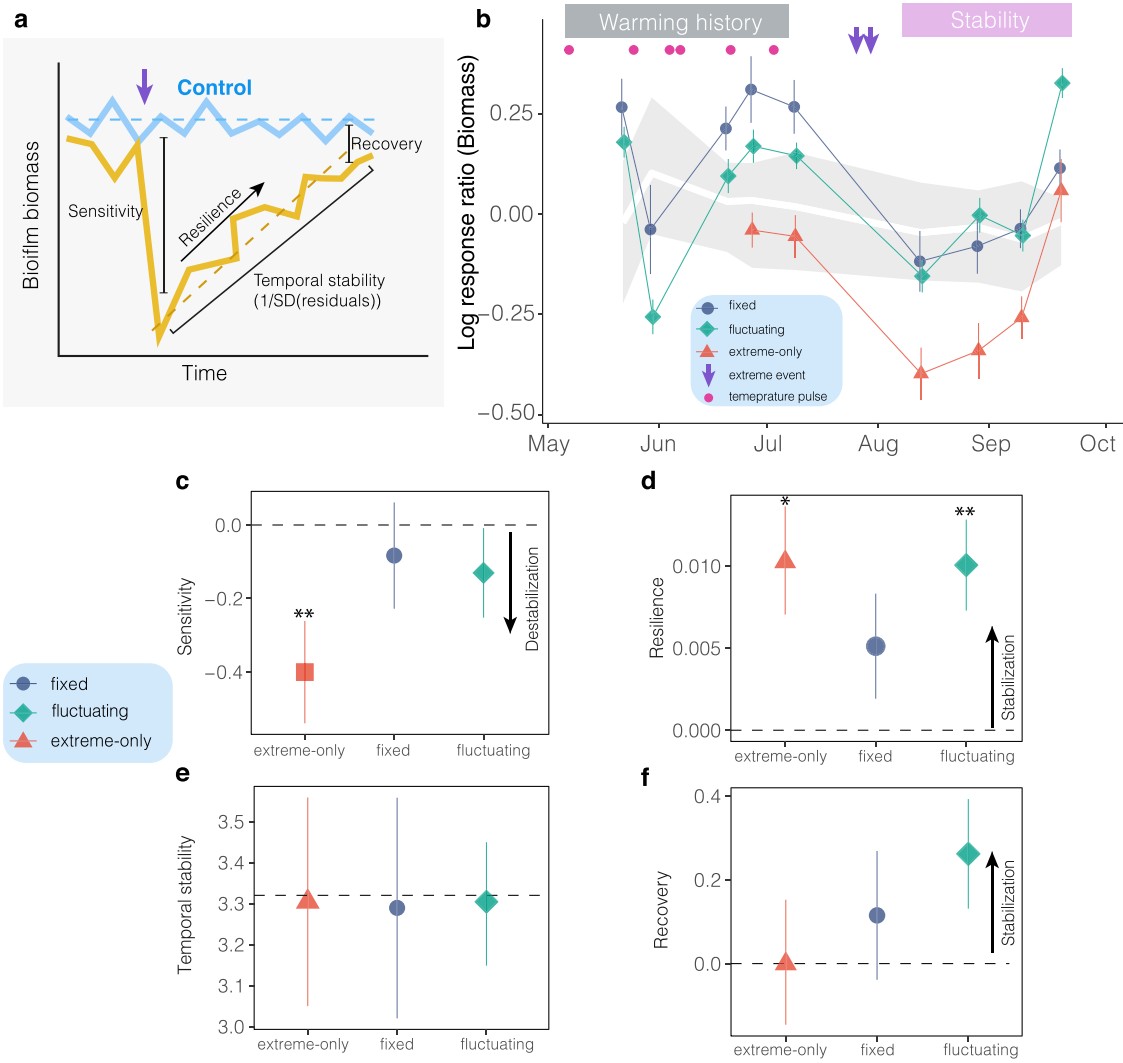

**Fig. 6 | Fixed and fluctuating warming stabilize biofilm communities against extreme temperatures. a** Graphic representation of the four stability components analyzed − sensitivity, resilience, temporal stability, and recovery − based on the response of treated (light orange) plots relative to controls (cyan). Sensitivity reflects the deviation of treated plots from controls at the first sampling after the imposition of extreme temperatures. Resilience is estimated as the difference in regression slopes between treatments and controls. Temporal stability is computed as the inverse of the standard deviation of residuals from linear mixed-effects models, and recovery is measured as the deviation between treatments and controls at the end of the experiment. **b** Mean log-response ratio (i.e., log-response of biomass in treated versus control plots) for different temporal warming regimes (mean ± SE). The control condition is represented with a bias-corrected and accelerated 95% confidence interval (BCa 95% CI) in light grey. These values and

their uncertainty metrics (SE and BCa 95% CI) were calculated from 24 observations (six subplots within each of four plots per condition). Pink symbols indicate temperature pulses, while purple down-facing arrows denote extreme temperature events. Pink symbols indicate temperature pulses, while purple down-facing arrows indicate the extreme temperature events. Multiple stability dimensions: (**c**) sensitivity, (**d**) resilience, (**e**) temporal stability, and (**f**), recovery in response to warming regimes. Points represent estimates of stability-dimensions from linear mixed effect model, and error bars indicate bootstrapped SE ($n = 1000$). $p < 0.05$, **$p < 0.01$, *** $p < 0.001$; Wald $z$-tests for fixed effects with p-values adjusted for multiple comparisons at the last sampling date using the false discovery rate (FDR) method. All statistical values are provided in Supplementary Tables 11, available in the Source Data. Source data are provided as a Source Data file.

## Experimental design

To assess how different levels of warming fluctuations affected the response of biofilm to subsequent extreme events, we imposed two temporal regimes of warming: 1) fixed, where we imposed pulses of temperature of the same magnitude (12 °C above ambient temperature), 2) fluctuating, in which temperature varied at each pulse application, producing a scenario with the same mean temperature as the fixed-warming scenario (12 °C above ambient temperature), but with a different variance (SD = 5). To generalize the effect of the fluctuating warming treatment (i.e., not making it contingent on one specific sequence of events), we established three different temporal warming sequences (fluct-s1, fluct-s2, fluct-s3) that had the same mean (ΔT = 12 °C) and variance (SD = 5) but different temporal profiles. Warming

manipulations were performed six times at a biweekly frequency from May to August 2018. The warming session consisted of maintaining for 70 min the difference between the warming chamber and the ambient air temperature as close as possible to a nominal treatment level (e.g., +12 °C). The warming device consisted of an aluminum box equipped with an adjustable butane stove and an opening to regulate the heating (Supplementary Fig. 1b). According to previous studies[49,50], an increase of 12 °C relative to the ambient temperature was expected to reduce biofilm biomass without completely eradicating biofilm. According to study-site data, warming pulses represent moderately recurrent thermal events typical of midday hours, especially during low tide and periods of high barometric pressure (Fig. 1c). Our aim was not to strictly mimic temperature predictions from regional models but

rather to simulate an increase in temporal variability, which is likely to become increasingly prevalent in the Mediterranean area due to ongoing climate change[51–53].

To assess the stability of biofilm, after the first series of warming treatments (phase 1 of the experiment), an extreme thermal perturbation was applied in which the biofilm was exposed to a temperature of 60 °C for 120 min twice in four days (phase 2). This was a strong, but realistic perturbation, as temperature peaks of 60 °C for 2 h in sunny conditions have been occasionally recorded at the study site[49,51] (Fig. 1c). To determine the effect of extreme events, we established four additional plots that received only the extreme temperatures and were therefore not previously disturbed (Extreme-Only). The warming device consisted of an aluminum box equipped with an adjustable butane stove and an opening to regulate the heating (Supplementary Fig. 1b). Aerial temperature was constantly measured with iButton loggers during the warming sessions inside and outside the heating chambers. Experimental plots were located 2–8 m apart and consisted of portions of substratum of 40 × 40 cm marked at their corners with rawlplugs inserted into the rock for subsequent relocation. Four experimental plots were randomly assigned to experimental conditions and interspersed across the study area. Four unmanipulated plots were used as controls. Three additional plots were established as control for artifacts (CA) to assess the potential effect of shading on biofilm during the heating sessions. CA plots were shaded with cardboard chambers of the same size of warming ones, but they were not warmed.

## Data collection
Biofilm biomass was determined by means of an image-based remote sensing technique that uses chlorophyll a concentration as a proxy for biofilm biomass (μg chl $a$ cm$^{-2}$)[20]. Chlorophyll $a$ was estimated from a ratio of reflectance at near-infrared (NIR) and red bands (Ratio Vegetation Index - RVI) by means of a IR-sensitive camera (ADC)[19]. The exposure time of the camera was set according to the light conditions that existed at the moment of image acquisition. Because light conditions varied during sampling, a reflectance standard (30% reflective Spectralon®), with a Lambertian surface, was placed within the field of view of the camera. NIR/red ratios were related to chlorophyll $a$ by a linear relationship, calculated based on of laboratory chlorophyll $a$ extraction from Calafuria sandstone cores[20]. For each photo of a plot, six subplots of 128 × 128 pixels were haphazardly taken and then processed with a Java routine on ImageJ software to obtain biofilm biomass estimates for each pixel. The spatial resolution (i.e., the area of each pixel) was 0.2 mm$^2$. Biofilm biomass was evaluated in nine times during the duration of the study, except for Extreme-Only plots, which were sampled starting from mid-June, 40 days before the imposition of temperature extremes. Notably, we ensured that subplots were spaced at a minimum distance of 30 cm from one another to minimize the likelihood of reselecting the same subplot on different sampling dates. Subplots were pooled across replicates and used exclusively for visualization purposes in Fig. 6b, while all statistical analyses were conducted using replicate-level data (plot) (see below).

## Whole-metagenome sequencing and assembly of metagenome assemble genomes
We collected surface rock samples (4 × 4 cm) covered in biofilm with a hammer and chisel from three plots per treatment at each of four time points: at the beginning of the experiment, after the imposition of thermal history, after the extreme events, and at the end of the experiment. Samples were taken haphazardly, leaving a buffer area of 5 cm from the edges of neighboring samples. A temporal lag of 5 to 7 days between treatment application and sample collection allowed the biofilm to adequately respond to thermal perturbations. Rock samples were kept on ice and in dark conditions for less than two hours before being stored at −80 °C. Prior to DNA extraction, we manually

removed the excess rock to increase the biofilm/rock ratio and maximize the yield of the DNA extraction. The rock samples were then pulverized with a mortar and pestle, and a 0.5 g sample was taken for DNA extraction through the Qiagen DNeasy PowerSoil Kit according to the manufacturer's instructions. DNA concentration and quality were determined using both NanoDrop ND-1000 spectrophotometer and Qubit Fluorometric Quantitation (DS DNA High-Sensitivity Kit, ThermoFisher). DNA sequencing libraries were prepared at IGA Technology Services Srl (Udine, Italy) and then sequenced using an Illumina HiSeq2500 sequencer (Illumina, San Diego, CA) in a 2 × 150 bp paired-end mode, generating a total of 959,137,528 reads, with an average of 12,456,332 reads per sample. Reads were de-multiplexed based on Illumina indexing system. One sample (fixed warming at the initial stage of the experiment) could not be sequenced due to an extremely low DNA concentration, resulting in a total of 77 samples available for analysis.

To reconstruct the Metagenome-Assembled Genome (MAG), we used the MetaWRAP pipeline v1.3.2[54]. Using the *read_qc* module with default parameters, raw reads were quality-trimmed, and human contamination was removed. Only high-quality reads from each sample were individually assembled into contigs through MEGAHIT (v1.1.2)[55] with a minimum contig length of 1000 bp; the quality of these assemblies was evaluated using QUAST (v5.0.2)[56]. High-quality contigs were grouped into Metagenome-Assembled Genome (MAGs) using the maxbin2, metabat2, and concoct algorithms included in MetaWRAP[54]. The output from these three algorithms was then integrated and refined into an optimized set of MAGs from a single assembly. As a result, 137 MAGs were generated, and their quality was assessed using CheckM (v1.0.18)[57]. Finally, the relative abundance of each MAG in each sample was calculated as the length-weighted average of the contigs in each MAG and expressed as Transcripts per Million (TPM).

Taxonomic annotation of MAGs was performed using GTDB-Tk v1.3.0, which is based on GTDB ver 95[58]. PhyloPhlAn 3.0 was used to create a maximum-likelihood phylogenetic tree for 136 MAGs[59]. One MAG (mag_18) was excluded from the analysis due to excessive fragmentation, which led to poor alignment and the detection of fewer than 50 marker genes, making its phylogenetic placement unreliable.

To address the potential underestimation of rare taxa in MAG reconstruction and better characterize the taxonomic diversity of biofilm, we first extracted 16S rRNA gene sequences from the quality-trimmed reads using SortMeRNA v4.2[60], with the SILVA ver. 128.2[61] reference database, which includes both bacterial and archaeal sequences. These reads were then assembled using SPAdes v3.15.2[62] in metagenomics mode, tailored for the diversity and complexity of metagenomic datasets, yielding a set of contigs representing reconstructed 16S rRNA gene sequences. We aligned quality-trimmed paired-end reads from each sample back to these 16S contigs using Bowtie2 v2.4.4[63] with default settings. The abundance of each contig was quantified by counting the aligned reads, with read counts normalized to TPM to account for variations in contig length and sequencing depth across samples.

## Functional profiling
We used a novel analytical framework (*microTrait*)[64,65] to map and translate microbial genomes into a hierarchical trait space that included energy acquisition, resource acquisition, stress tolerance, and life history traits that underlie different microbial strategies[66]. This analysis involved two main steps: 1) genome sequences were converted into open reading frames using Prodigal[67] the resulting protein sequences were then scanned through a library of gene-level Hidden Markov Models (*microTrait*-HMMs, with HMMER/hmmsearch from http://hmmer.org) to produce a count table for the detected gene modules. These procedures generated trait matrices (MAG × ecological traits) at different resolutions corresponding to the levels of the functional hierarchy. *MicroTrait* rules mapped protein family content

into traits using Boolean logic. Each protein family is represented as a Boolean variable (equals to 1 if detected, 0 otherwise) whose value was determined by the output of the corresponding microTrait-HMM. Cross-references to KEGG orthologs (KO, from https://www.genome.jp/kegg-bin/get_htext) (when available) were used to determine the cut-offs.

The results of microTrait analysis should be interpreted cautiously as they represent the potential functions of a microbial community. microTrait established strict gene-level microTraitHMM thresholds based on KEGG Orthology (KO) group genes (with an accuracy greater than 99.99%), which, combined with high levels of genome completeness, resulted in highly accurate trait assignment. Additionally, it focuses on mechanistically well-studied traits with documented genetic bases, providing more accurate functional trait characterization compared to methods like Picrust2, which relies on predicted metagenomes derived from Amplicon Sequencing Variants (ASVs).

The minimal doubling time (MDT; hours) of each MAG was calculated from codon usage bias patterns in each MAG with >10 ribosomal proteins using the R package "gRodon"[26,68].

## Data analysis

**Taxonomic and functional diversity.** Taxonomic diversity was estimated as the equivalent number of species by calculating the first three Hill numbers using the hillR package: MAG richness ($q = 0$), the exponential of Shannon's entropy $(q = 1$, referring to Shannon diversity), and the inverse of Simpson's concentration ($q = 2$, referring to Simpson diversity)[69]. Spatial synchrony was calculated for each plot as the average population synchrony across MAGs, using Gross' η as the synchrony metric, weighted by their relative abundance[70,71]. Population variability was obtained as the weighted mean of the standard deviation of the TPM of each MAG for each plot[45]. Spatial synchrony and population variability were calculated using the tempo and codyn R packages, respectively.

The contribution of stochastic and deterministic processes underlying the assembly of the biofilm community was quantified through a null-model analysis using the phylogenetic normalized stochasticity ratio (pNST). pNST was based on the β-nearest taxon index (βNTI)[72]. Briefly, the analysis compares the observed phylogenetic dissimilarity (β-nearest taxon index [βNTI])[72] with that expected under null conditions[73]. pNST metric ranges from 0, representing absolute determinism, to 100%, representing complete stochasticity. Statistical comparisons between treatments at the 2nd, 3rd, and 4th sampling dates were performed using a bootstrapping approach, with each dataset resampled 1000 times. *P*-values were adjusted for multiple comparisons using the false discovery rate (FDR) procedure, based on the number of comparisons at each sampling date. The analyses were performed with the *pNST* and *nst.boot* functions within the NST R package[24].

Since one of our primary goals was to assess the functional shifts underlying the response of biofilm to warming regimes, the calculation of functional entities was based on stress tolerance traits. Functional group richness (FR) was calculated as the number of functional entities (FE), which represent the number of MAGs sharing the same combination of stress tolerance traits. Because functional redundancy calculated as the ratio between the number of MAGs and FE may be biased, we computed functional over-redundancy (FOR) as proposed by Mouillot et al.[25]. FOR represents the tendency of most MAGs to clump into a few highly abundant functional entities. FOR ranges from 0 to 1; low values of FOR indicate that most FEs contain the same number of MAGs, and high values indicate that most MAGs are contained in the richest FE. Finally, we computed functional vulnerability, which is the proportion of FEs represented by a single MAG. FEs and functional metrics were computed using the *sp.to.fe* and *alpha.fd.fe* functions, respectively, in the R package mFD[74]. We used the number of stress-related gene modules in each MAG as a proxy for stress tolerance.

**Statistical analysis.** We used Linear Mixed Effects Models (LMEMs) to assess the effect of different temporal regimes of warming on taxonomic and functional diversity[75]. The fixed part of the model included the "Temporal Regime of Warming" (5 levels: controls, fixed warming, fluct-s1+ fluct-s2+ fluct-s3) and "Time" (number days). To test the effect of fluctuating warming, "Temporal Regime of Warming" was partitioned into two planned contrasts: Control *versus* Fluctuating (Control *vs* Fluct. [fluct-s1+ fluct-s2+ fluct-s3] and Fixed *versus* Fluctuating (Fixed *vs* Fluct. [fluct-s1+ fluct-s2+ fluct-s3]). Each term and contrast were crossed with "Time". The random part of the model included the variance in intercepts and slopes among plots (*Time||plot*). Since the main aim of the study was to assess the effects of varying warming regimes, the analysis involved the 2nd, 3rd, and 4th sampling date, for which treated plots received the same amount of warming. The analysis was centered on the 2nd sampling date so that deviations from the intercept (control) reflected the effects of the history of warming. In addition, to assess how warming history affected the recovery of biofilm following extreme temperatures, we performed contrasts equivalent to those of the main analysis, centered on the last sampling day. LMEM and contrasts were conducted using the *glmmTMB* and *emmeans* functions from the "glmmTMB" and "emmeans" R packages, respectively. *P*-values for contrasts at the last sampling date were adjusted using the false discovery rate (FDR) method. Model assumptions were assessed visually using plots of residuals *vs* fitted values, box plots of residuals *vs.* experimental conditions, and QQ plots of standardized residuals *vs* Normal quantiles generated with the "performance" R package[76].

A differential abundance analysis of MAG counts in fluctuating and fixed warming treatments relative to controls following temperature extremes was performed with the DESeq2 R package[77]. To assess the trade-off between stress tolerance and growth rate, we examined the relationship between the number of gene modules related to stress-tolerance traits in each MAG and the minimal doubling time. Since the number of gene modules may also be influenced by genome size and MAG completeness, we included these variables as covariates in the model. Additionally, to account for variance in MDT, which changes with genome size, we applied a dispersion model (~log(genome_size)).

We also conducted an analysis to assess differences between control and extreme-only plots during the first phase of the experiment, before the application of extreme temperatures. The fixed effects included Treatment, Time, and their interaction, while the random effect was limited to plot ID (1|plot_id) to avoid convergence issues that arose when including Time as random slope.

**Stability components.** For each treatment, we computed four components of stability (sensitivity, resilience, temporal stability, and recovery) separately for biomass, dark and light yield using LMEMs[28,78]. Treatment and time (number of days from the application of temperature extremes) were included in the fixed part of the model. Plot identity was included the random part. Time was also used as a covariate in the random part of the model to account for the lack of temporal independence of repeated observations within plots (*Time||plot*). Stability components were assessed over the five sampling dates following the application of temperature extremes pulses (from the 6th to the last sampling date). Controls were set as the baseline (Intercept) such that main effects (deviations between the Control and the treatments) reflected sensitivity after the imposition of extreme temperatures. Linear deviations between treatments and Control (treatment × time interactions) were used to quantify resilience. The following linear model was used:

$$y_{i,j,k} = \alpha_0 + \alpha_{0i} + \beta_1(Time) + \beta_{1i}(Time * Treat_i) + a_{0k} + b_{1j} + \varepsilon_{i,j} \quad (1)$$

Where $y$ is the response variable (biofilm biomass) in the $k$-th plot and in the $i$-th treatment, $\alpha_0$ and $\beta_1$ are the intercept – centered at the first date after the imposition of extreme events – and the slope of controls; $\alpha_i$ is the main effect (sensitivity) of the $i$-th treatment, $\beta_{1i}$ is the deviation from the linear trend of the $i$-th treatment (resilience), $a_{0j}$ and $b_{1j}$ are the random effect intercept and slope of the $j$-th plot and $\varepsilon_{i,j}$ is the residual error.

Recovery was then estimated as the deviation between controls and each treatment at the last sampling date, using predicted values generated from LMEM (Eq. 1). Finally, temporal stability was quantified as the inverse of the standard deviation of the residuals. To derive measures of uncertainty for effects estimated through LMEM, we computed 1000 parametric bootstraps of each fitted model. Errors bars of sensitivity, resilience, temporal stability, and recovery are presented as the 95% confidence intervals (CIs).

To more accurately determine how a history of warming influences biofilm responses to extreme temperature events, we conducted a Before-After Control Impact (BACI) analysis. This analysis focused on two sampling dates before and after the temperature extremes. In our model, the fixed component encompassed both the treatment and the Before-After condition, as well as their interaction. The random component included the plot ID (1|plot) and the Before-After condition, which was nested within the sampling date (Before-After:Time).

To control for potential artifacts, we conducted a separate analysis where the treatment was the sole covariate in the fixed part of the model. Here, the random structure included only the plot ID (1|plot).

### Reporting summary

Further information on research design is available in the Nature Portfolio Reporting Summary linked to this article.

### Data availability

All sequencing data are available in the NCBI BioProject PRJNA1062819 at: https://www.ncbi.nlm.nih.gov/bioproject/PRJNA1062819. All other relevant data generated or analyzed during this study are included in the manuscript and supplementary information. Source data are available for Figs. 1c, d, 2, 3, 4, 5, 6b–d, and Supplementary Figs. 3–9 in the associated source data file. Source data are provided with this paper. Data that support the findings of this study can also be accessed on Figshare (https://doi.org/10.6084/m9.figshare.28319627.v1). Source data are provided with this paper.

### Code availability

All R scripts used for data analysis and figure generation are publicly available on Figshare (https://doi.org/10.6084/m9.figshare.28319627.v1).

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

## Acknowledgements

This research was partially funded by ACTNOW "Advancing understanding of Cumulative Impacts on European marine biodiversity, ecosystem functions and services for human wellbeing" (Horizon Project No 101060072) and by the University of Pisa grant PRA_2020_76 awarded to L.B.C. M.M. extends gratitude to Alan Barozzi for the insightful discussions on metagenomic analysis.

## Author contributions

L.B.C. and L.R. conceived and designed the experiment. J.H. and L.R. performed the field experiment and sampling. M.M. and J.H. performed the laboratory analysis. M.M., J.H., and L.R. carried out the bioinformatic analyses of genomic data. L.R. performed the statistical analysis. M.D.A., E.P. and M.M. contributed essentially to establishing the genomic pipeline, overseeing data generation, and supporting its interpretation. L.R. wrote the first draft of the manuscript. L.B.C. and M.D.A. were also involved in the critical revision of the manuscript. All the authors commented on the manuscript.

## Competing interests

The authors declare no competing interests.
