## [Transparent Peer Review file · Nature Communications]

Legacies of temperature fluctuations promote stability in marine biofilm communities

Corresponding Author: Dr Luca Rindi

A version of this paper was originally rejected for publication by Nature Communications, however that decision was reconsidered after appeal by the authors.

Version 0:

Reviewer comments:

Reviewer #1

(Remarks to the Author)

Rindi and colleagues present the results of an interesting study on temperature regimes and legacy effects on stability and diversity of biofilms. I commend the authors on their hard work with this experiment, but have some concerns regarding experimental design and/or interpretation which leave me not wholly convinced that the results of the study can be interested as here. I preface this review with a note that my background is in stability, including mesocosm experiments, not specifically with the microbial/genetic components of the manuscript, so I mostly do not comment on that.

I am a bit unsure about how the temperature treatments are classified and presented. Given that extreme temperatures are typically defined as those outside of the 95% percentiles (and heatwaves, for instance are periods of 3+ days outside those temperatures), the pulses themselves are also extreme temperatures. So defining an extreme event separately to these pulses seems a bit confusing.

Similarly, how were +12 degrees and pulses of 60 degrees chosen? Are these realistic temperature scenarios (I do not know about the specific study system but 60 degrees sounds very extreme to me – I note later the authors say 55 has been observed. It may be worth making that clearer)? I note later the authors say these were meant to “simulate warming scenarios that may become increasingly prevalent” but I have no idea about the climate projections for this region; is this likely to be a relevant manipulation based on local future projections?

Measuring responses of the extreme-only treatment only after the extreme event makes it difficult to say for certain whether differences between extreme-only and other treatments are meaningful. For example, could stochasticity ratios, functional diversity, or other variables have converged on/diverged from other treatments only after the extreme events? I think this is a question the authors are trying to ask with this treatment, but without explicitly presenting pre-disturbance data from this treatment, we cannot rule out the possibility that any effects were pre-disturbance effects.

I am a bit unclear on the sample size used here in terms of the number of plots (the unit of measurement). The methods section hints that it may be 4 plots per treatment, but I am not sure. If so, this is pushing the limit of a meaningful comparison between treatments due to low sample size. I would be reluctant to make broad conclusions based on data from only 4 plots per treatment, but perhaps I misunderstood.

Specific comments

L75: it is not clear from this line how environmental fluctuations relax competition.

L99-108: I understand that these are two competing hypotheses, but it wasn't clear to me which the authors aimed to test and how from this paragraph. E.g. there's discussion of stress tolerance vs growth ability, but this cannot be inferred by simply measuring functional diversity. So did the authors aim to identify stress tolerant or vulnerable taxa? Please make the specific aims of the study clearer in this paragraph.

L122: were not all treatments confronted with successive warming events (there are two points on Figure 1 for all treatments)? This sentence confused me.

L161: my understanding of the experimental design is that this experiment was conducted on pre-assembled biofilms? In which case, it may be misleading to refer to the effects of warming on reassembly; priority effects etc. may have determined competitive outcomes during the original pre-experiment assembly process. Instead I think these analyses represent treatment effects on composition.

L219: I think susceptibility should be "sensitivity" here. The other dimensions of stability did not significantly differ.

L218-237: I am confused here because the authors are referring to controls, but I think we're supposed to be thinking about the "constant" warming scenario. Please be careful about language usage here because control treatments were established earlier in the paper as being something else.

L249: missing word?

L306: see comment above, I wouldn't describe this as "constant warming".

L335: Why was 70 minutes chosen? This seems a remarkably short exposure time to cause the kinds of treatment-level differences presented in the manuscript. I am quite surprised there were effects at all. After how long were measurements then taken? Was it immediately after the 70-minute heating? If so, that could explain how treatment effects emerged despite the very short heat pulses.

L418: I find the explanation of the synchrony calculation to not be well enough explained. Please expand.

L431: I don't think this is really functional richness, but perhaps would be better defined as "functional group richness" since it is based on functional groupings. See the literature on functional groups for more info.

Fig 1: I'm having a hard time understand the definition of constant warming used here. The authors seem not to be referring to a constant warming press perturbation, but rather pulse perturbations to the same temperature (have I understood correctly)? Calling this "constant warming" I think is a bit misleading as actually communities under a constant warming treatment are in this case no more frequently exposed to warmer temperatures than those under fluctuating temperatures. Perhaps a workaround would be to call these "fixed-warming heatwaves" or something similar, to make much clearer what these warming events actually represent.

I also generally found figure 1 to be confusing, I did not follow the reasoning as written in the (wordy) blue boxes, and therefore was not convinced of why we would expect stability responses to as suggested. I wonder if replacing these with a more concrete example may help?

Fig 1a: "history", "ambient temperature", "reduce", "resistance" are all misspelled.

Fig. 2: Were the control plots also exposed to extreme events? If not, the dip in panels a and b following the extreme events is interesting and perhaps worth discussing (some external environmental effect perhaps?). Also "temperature" is misspelled.

Fig. 2c,d: could the authors denote significant pairwise contrasts on the figure (e.g. using letters or * or similar)? This would improve the clarity of the figure.

Fig. 3: "Temperature" is misspelled.

Fig. 5 and stability components: It may be helpful to readers unfamiliar with the specific stability components in the analysis if stability components were to all represent the same direction of interpretation. By which I mean e.g. high sensitivity = low stability, but high resilience = high stability. Could some of the axes be reversed or labels added to aid interpretation?

Reviewer #2

(Remarks to the Author)

In the manuscript entitled "Legacies of temperature fluctuations promote stability in marine biofilm communities", the authors exposed microbial communities from biofilms in a rocky intertidal environment to different treatments of temperature to compare the effects of fluctuating vs. rather constant warming history on the response of the microbial communities toward extreme warming events. Overall, the study necessitated a lot of work, and presents very interesting results. The novel approach is well-designed, and the analyses are conducted carefully. I think that the results are of great significance to the field. My principal concern regards the structure of the manuscript. As the Results are described before the Material and Methods, some parts of the results feel more like M&M for me. I understand that these parts are necessary to understand the following results, but it makes the structure of the manuscript a little bit difficult to follow. As for now, the Results section is not only results, but presents also a lot of Materials and Methods, and even some part that are repetitions of the Introduction. I am not sure if it is possible to change the structure of the manuscript, but, in my opinion, putting the Material and Method section before the Results and Discussion would greatly help gaining readability and understandability of the manuscript. Finally, I also have some specific comments that I listed hereafter.

L48. «of several ecosystems”. Maybe this part could be changed to “of many ecosystems”. Indeed, plenty of ecosystems were already shown to be affected by extreme climate events.

L88. “fluctuation warming conditions”. This should be change to “fluctuating warming conditions”.

L128. “138 MAGs”. You should define MAG the first time the abbreviation is being used.

L133. Maybe the name of the test being used here (Linear Mixed Effect Model) should be written in the parenthesis, at least the first time some results of such tests are described.

L152-154. This is a repetition to what was already presented in the Introduction section.

L154-161. Similarly, this part does not belong to the Results section, but rather to M&M.

L166-168. Again, this belongs to the Discussion section, not results.

L174-176. This is a repetition of the hypotheses presented in the Introduction section.

L195-196. This does not belong to Results, rather to either Introduction or Discussion sections.

L249. “our research revealed played a crucial role in augmenting biomass”. A word is missing, please reformulate.

L254. “ High fluctuating warming” should be replaced by “Highly fluctuating warming”.

L158-260. I agree, this history is far too often neglected in current research.

L307. “On the other end” should be replaced by “On the other hand”. In addition, this must be used with “on the one hand” first.

L308. Should “susceptibility” be replaced by “sensitivity”?

L309-311. Keep in mind that you only studied microbial communities from a rocky intertidal area. While the results are very interesting, any extrapolation of the results should be done with caution, as microbial communities from different environment, and therefore with different community composition, may respond in another way to warming and unpredictability.

L335. Why 70 minutes? It is a very short duration, compared to what can occur in natural environments.

L336. What chambers? Please be more precise in the description of how the warming treatments were performed on the biofilms. Did the chambers let light go through during the warming treatments?

L326-343. How did you make sure that all plots (the warmed and the control ones) were similar in composition at the beginning of the experiment?

L360. Why are the references not in the same format as the others?

L415-420. I believe these analyses were performed with R?

L434. “Muillot et al.” should be replaced by “Mouillot et al.”.

L725 and L734. I would replace “depauperated” by “less diverse”.

L736. “indicate” should be “indicates”.

L753, L760, L770. Please refer here to the fluctuating sequences as “fluct-s1”, “fluct-s2”, and “fluct-s3”, to be consistent with other parts of the manuscript.

L771. “and is expressed as community-level weighted mean”. The “as” is missing.

Reviewer #3

(Remarks to the Author)

The ms by Rindi et al. uses microbial communities that form biofilms on intertidal rocky shores to experimentally examine effects of constant and fluctuating temperature stress on microbial community diversity and function, and how their responses to these different stressors affect overall responses to an extreme temperature event, with particular focus on stability (resistance, resilience and temporal stability) of biofilm biomass.

This work is timely as although there has been significant research on the effect of different types of temperature stress (or other constant vs variable stressors) on ecological communities, how these effects then affect subsequent responses to stressors has been largely overlooked. This is important because such information is needed in order to better predict the fate of biodiversity and ecosystem function under future environmental change.

Importantly, this work quantitatively examine key aspects of stability of communities: resistance to stress, resilience and temporal stability, and uses information of diversity (who are they?) and function(what can they do?) of multiple members in the microbial community to test hypotheses about functional redundancy and other mechanisms which may explain overall stability in the context of stress.

Besides the rationale, importance and easy of manipulation of the study system and the environmental context (e.g. temperatures based on realistic predictions), there are two key aspects that make this experiment very relevant: (i) it is done in the field, in the context of natural communities and natural variation in factors other than those manipulated here; (ii) the experimental design is appropriate and rigorous (e.g. replication of fluctuating treatments, procedural controls), enabling rigorous tests of the proposed hypotheses.

The main issue with this work is the data used to characterise microbial community diversity and (potential) function. The authors used metagenomic sequencing to then assemble metagenome-assembled genomes (MAGs), which resulted in 138 MAGs, or microbial “species” for which they also have information on their potential functional role – ‘potential’ because one does not know whether the genes observed in their genome are being expressed or not, as no RNA/gene expression was quantified in this study. A key question is whether those 138 MAGs are representative of the diversity and functional potential of the entire microbial community. These communities typically have many hundreds to thousands of taxa (ASVs characterised by e.g. 16S rRNA gene amplicon sequencing) and functional genes (via metagenomics). While MAGs allows understanding which functional genes are present in which taxa, and thus better tackle the functional redundancy issue, this study should have also used information of taxonomic and functional gene diversity to test hypotheses about changes in

diversity and function in response to treatments. For example, would the patterns observed in richness or other diversity measures remain if data of the entire community (eg via 16S sequencing) were used?

A second important issue has to do with the ecological trait assignment to MAGs/genes. How accurate is this for this system? These assignments are based on databases which tend to be system-specific and a lot of understanding of particular taxa, their functions and roles in that system are needed to accurately predict/assign ecological traits. If the ecological traits assigned are not accurate, then this will have implications for the observed functional responses and the interpretation of the data. This is more problematic where actual microbial function (eg via gene expression or proteomics/metabolomics) has not been measured to verify whether assigned traits make sense.

I think the authors need to address the two major points I raise above in the Discussion. Perhaps also the structure of the ms could be modified so that biofilm stability is presented first and then analyses of microbial diversity and function come after as potential mechanisms that may explain effects on biofilm stability. I also suggest revising the Introduction as some parts are a bit repetitive.

Minor comments

Lines 67-70: Sentence too long. Suggest ending after 'natural communities' and deleting the rest as it's not clear.

L 78-81: It is worth mentioning here that there could also be trade-offs among stressors, i.e. that traits that lead to resistance to one stressor (stressor 'A') may lead to lower resistance to another stressor ('B'). In the context of multiple-stressors this is a very important point which may impact diversity/stability relationships.

L 83-86: repetitive from above (eg 71-73).

L 88: "fluctuating"

L 120: include SD, state these were multiple fluctuating treatments and refer to Methods for the rationale

L 128: were there 138 or 137 MAGs as per lines L 392, 397?

L151: effect "of" fluctuating temperatures on...

Version 1:

Reviewer comments:

Reviewer #1

(Remarks to the Author)

I previously reviewed this manuscript by Rindi et al. as reviewer 1 and find this version much improved. I appreciated their detailed and thorough response to reviewer comments, and now have only minor comments below. I do think the figures could be further modified to clarify that the extreme-event-only treatment didn't differ from control before manipulation (as shown in the response letter). Whether in the main manuscript or at least in the supplement, this further addition would be useful to many readers I think.

Specific comments:

L63: what does "diversified" mean here? I think the authors might be referring to the concept of response diversity – a wider variety of different responses to environmental change – but am not sure. Or do you simply mean biodiversity (species richness)?

L138-141: If I understand this correctly, this addresses my previous comment about only considering the extreme event treatments after the events. In your response to reviewers, you offered to include the figures in the manuscript. Indeed I think that would be useful here (in the supplement) and perhaps also mapped onto Fig. 1c.

L178-179: sorry if I missed it but what is "compositional diversity"? Do you mean compositional dissimilarity (which would be beta diversity). As far as I've always understood the concept, composition doesn't have diversity – unless we're talking about diversity in space (between patches)?

L212: maybe worth citing the Hillebrand et al paper here to clarify what you mean.

L240: "taxonomic" I think

L314: "rather than reducing stability" seems like a separate idea here. Is the opposite of constant thermal warming a reduction in stability? Which dimension of stability are we talking about? Consider rephrasing this sentence

L315: I don't know if I agree that warming can "provide a broader niche space". Warming is by definition just shifting the position along an environmental axis, and so will benefit species with their thermal optima, while disadvantaging others. Please consider rephrasing this whole sentence or clarifying these ideas.

I noticed that Dal Bello et al is cited as two different references. There may be other similar issues, but I did not check in detail.

Reviewer #2

(Remarks to the Author)

I want to thank the authors for having taken the time to carefully answer all my comments on the previous version of the manuscript. I think that the manuscript is clearer now, notably why and how the experimental manipulations were implemented is now much clearer.

Reviewer #3

(Remarks to the Author)

The authors have done a good job at addressing my (and other Reviewer's) comments. I think the inclusion of analyses of the 16S rRNA gene contigs provides more robustness on the interpretation of the data. My main comment is that there should be 1 sentence in the Discussion that states, up front, that when referring to a community in your inferences – particularly function (as the inclusion of the 16S data now captures most of the taxonomic composition) – you are focusing on dominant taxa obtained via MAGs.

Some minor comments below:

Reviewer 1 made an interesting comment with regards to presenting data from the extreme plots before the application of the treatment. I agree with the response from the authors but I think it would be useful to include these (in Supplementary?) data given that the authors have it and have already analysed it in their response to the Reviewer's comment.

Line 101: to 'the' biofilm community.

L 154 and throughout: check that the terminology used now – 'fixed' – has been consistently changed as some contrasts and statistical tables still refer to 'const'.

L 160: rare what? taxa?

L 241: here and throughout the Discussion, rather than inferring patterns for the entire community, specify that the focus was on some, likely key members of the community. E.g. "...functional structure of dominant members of a community."

L 416: join these two sentences.

L 807: add 'inter-specific' before competition.

Supplementary Fig. 6 b: x-axis should be 'fold' not 'old'.

Sup. Table S1 (and others): None of the terms in the model are significant except for the intercept – why do you report post-hoc tests between treatments at the last sampling time? This would be appropriate if there was a significant effect of treatment in an analysis of data only from that time point or if there was an interaction between treatment x time (where the other time points would also need to be included in these contrasts). However, you would need to run an analysis of variance testing overall effects of terms in the model first, rather than each term against the intercept as you have done here, e.g. using the Anova function in the R car package on this GLMM model (e.g. by fitting the model using the lmer function in the lme4 R package). I suggest you do this for all similar analyses.

Reviewers' comments are in italic, our replies are in normal text.

In main document the amendments are highlighted in red.

Reviewer #1 (Remarks to the Author):

“Rindi and colleagues present the results of an interesting study on temperature regimes and legacy effects on stability and diversity of biofilms. I commend the authors on their hard work with this experiment, but have some concerns regarding experimental design and/or interpretation which leave me not wholly convinced that the results of the study can be interested as here. I preface this review with a note that my background is in stability, including mesocosm experiments, not specifically with the microbial/genetic components of the manuscript, so I mostly do not comment on that.”

We thank the Reviewer for appreciating our experimental work. We have addressed the Reviewer's main concerns with data-driven responses. Specifically, 1) we used in-situ temperatures to accurately characterize our treatments in terms of probability of occurrence, 2) we provided both plots and analysis demonstrating no significant difference between control and extreme-only plots during the first phase of the experiment, and 3) we presented evidence of a trade-off between stress tolerance and growth ability in biofilm taxa, showing that this trade-off is maintained by fluctuating temperatures rather than fixed warming.

“I am a bit unsure about how the temperature treatments are classified and presented. Given that extreme temperatures are typically defined as those outside of the 95% percentiles (and heatwaves, for instance are periods of 3+ days outside those temperatures), the pulses themselves are also extreme temperatures. So defining an extreme event separately to these pulses seems a bit confusing.

Similarly, how were +12 degrees and pulses of 60 degrees chosen? Are these realistic temperature scenarios (I do not know about the specific study system but 60 degrees sounds very extreme to me – I note later the authors say 55 has been observed. It may be worth making that clearer)? I note later the authors say these were meant to “simulate warming scenarios that may become increasingly prevalent” but I have no idea about the climate projections for this region; is this likely to be a relevant manipulation based on local future projections?”

We agree with the Reviewer that the distinction between the two treatments may not have been as clear as intended in the original manuscript. To clarify, we utilized in-situ temperature data from previous studies to better characterize the thermal features of our study site (Fig. 1c). Our analysis indicates that temperature pulses 12°C above ambient levels occur with a probability of less than 20%, while extreme temperatures reaching 60°C have a probability of less than 1%. This data underscores that the two treatments represent distinct thermal scenarios, each with a clearly different level of rarity. Our intent was not to simulate treatments based on regional climate model predictions, as these would be poorly representative of local conditions. Instead, we aimed to construct treatments rooted in site-specific temperature data, capturing both increased climatic variability and extreme events pertinent to our study area. We have revised the manuscript to emphasize these distinctions more clearly (lines: 80-83, 316-324 and 327-329) and added an additional panel to Figure 1 (Fig. 1c and lines: 782-794), which illustrates the frequency distribution of in-situ temperatures, highlighting that our two treatments occupy distinct quantile categories.

“Measuring responses of the extreme-only treatment only after the extreme event makes it difficult to say for certain whether differences between extreme-only and other treatments are meaningful. For example, could stochasticity ratios, functional diversity, or other variables have converged on/diverged from other treatments only after the extreme events? I think this is a question the authors are trying to ask with this treatment, but

without explicitly presenting pre-disturbance data from this treatment, we cannot rule out the possibility that any effects were pre-disturbance effects.”

The Reviewer raised an important point. To address the Reviewer's concern, we conducted additional analyses comparing these only-extreme plots to the controls before the extreme temperature treatments (lines: 471-473) and generated figures (Figure 2 a,b, 3 and 4 included at the bottom this response) illustrating the separate trajectories for the controls and extreme-only conditions. The tests for all variables showed no significant differences (Supplementary Tables 7, 9 and 14). We initially chose not to include these figures in the manuscript because the primary aim was to compare the effects of extreme temperatures without prior warming exposure, which can be effectively achieved by comparing these two conditions after the extreme temperature imposition. In addition, the spatial random allocation the spatial interspersion of plots to different conditions ensures to distribute uncontrolled variability among experimental units therefore minimizing the likelihood to observe initial differences among experimental conditions. Given that the plots were randomly assigned to different experimental conditions, we expected that the controls and extreme-only plots would not exhibit any divergent patterns before the extreme temperature imposition. Our decision is also supported by the consistency of our graphical representations with the analyses, in which all unmanipulated plots in the first phase of the experiment were used as controls. This approach ensures clarity and alignment with our study's objectives. However, we are open to including these additional figures in the manuscript if the Editor or Reviewers believe they would enhance the clarity and informativeness of our findings.

Fig. 2

Fig. 3

Fig. 4

“I am a bit unclear on the sample size used here in terms of the number of plots (the unit of measurement). The methods section hints that it may be 4 plots per treatment, but I am not sure. If so, this is pushing the limit

of a meaningful comparison between treatments due to low sample size. I would be reluctant to make broad conclusions based on data from only 4 plots per treatment, but perhaps I misunderstood.”

The reviewer's point regarding the number of replicates is well-taken. Indeed, each experimental condition included four replicates. However, this should not detract from the validity of our results, as the outcomes of the analysis were statistically significant. It is important to note that a lack of significance, especially if the results were close to significant, might indicate a problem of limited statistical power due to a low number of replicates. However, this was not the case in our study. Our findings and conclusions are robustly supported by significant results, indicating that the experimental design and the number of replicates were adequately suited for the study's objectives. Furthermore, the decision on the number of replicates was informed by prior experimental studies conducted at the same study site. These studies, which also employed a similar number of replicates (3-4), consistently yielded significant results that substantiate our hypotheses (see Dal Bello et al. 2015, *Glob. Change Biol.*; Rindi et al. 2022, *Ecol. Evo.*; Rindi et al. 2022, *Mar. Environ. Res.*). Thus, we believe that our experimental design, including the number of replicates, is justified based on both the significant outcomes of our study and the precedent set by earlier research in similar settings.

“Specific comments”

“L75: it is not clear from this line how environmental fluctuations relax competition.”

Environmental fluctuations can relax competition through temporal niche differentiation, where different taxa are favored at different times. This process is expected to increase asynchrony across taxa, thereby reducing interspecific competition. The effect is particularly pronounced when there is substantial differentiation among taxa in their sensitivity to environmental conditions. As we streamlined the introduction (please refer to our response to Reviewer 3 below) this text has been eliminated from the current version of the manuscript.

“L99-108: I understand that these are two competing hypotheses, but it wasn't clear to me which the authors aimed to test and how from this paragraph. E.g. there's discussion of stress tolerance vs growth ability, but this cannot be inferred by simply measuring functional diversity. So did the authors aim to identify stress tolerant or vulnerable taxa? Please make the specific aims of the study clearer in this paragraph.”

These are indeed two competing, mutually exclusive hypotheses, each offering plausible predictions about the effects of past fluctuating temperatures on the functional and compositional structures of biofilm communities. As one hypothesis gains support from the data, the other becomes less plausible as an explanation for our findings. We agree with the Reviewer that our initial analysis was insufficient to directly test the trade-off between stress tolerance and growth ability. To address this, we quantified the stress tolerance of each MAG by counting the number of gene modules related to stress-tolerance traits and examining their relationship with minimal doubling time. Our analysis revealed a positive correlation between the number of stress-tolerance traits and minimal doubling time, indicating a trade-off between stress tolerance and growth within the biofilm community. This relationship persisted for MAGs that were more abundant under fluctuating conditions compared to controls, but it disappeared for MAGs more abundant under fixed warming conditions. These findings suggest that elevated warming fluctuations favor taxa constrained by the trade-off between stress tolerance and growth rate. This result enhances our understanding of how past environmental conditions shape the functional structure of biofilm communities. We have included a figure illustrating these findings and detailed the analysis (lines: 445-447 and 467-470) and results (Fig. 5 and lines: 165-175) in the manuscript and discussed this relationship (lines: 259-270).

“L122: were not all treatments confronted with successive warming events (there are two points on Figure 1 for all treatments)? This sentence confused me.”

The sentence was indeed unclear. Since both fluctuating and fixed warming plots received the extreme temperature, the extreme-only treatment served as a reference to discern the effect of past warming history from that of warming extremes. The two points in Figure 1 indicate the two subsequent extreme events. We have revised the sentence accordingly to clarify the meaning of the extreme-only condition (lines: 80-83).

“L161: my understanding of the experimental design is that this experiment was conducted on pre-assembled biofilms? In which case, it may be misleading to refer to the effects of warming on reassembly; priority effects etc. may have determined competitive outcomes during the original pre-experiment assembly process. Instead I think these analyses represent treatment effects on composition.”

We agree with the Reviewer that the analysis refers to the effect of treatments on the composition of biofilm. Following the Reviewer's suggestion, we have replaced the term “assembly” with “composition” (line: 87 and 118).

“L219: I think susceptibility should be “sensitivity” here. The other dimensions of stability did not significantly differ.”

We have replaced 'susceptibility' with 'sensitivity' (line 183). Additionally, the resilience (recovery rate) of both the extreme-only and fluctuating treatments significantly deviated from the controls, as detailed in lines 193-196 and Supplementary Table 16. Comparisons among treatments were made indirectly, based on whether one treatment produced a significant deviation from the control for a given component while another did not (please see also the response to the comment below).

“L218-237: I am confused here because the authors are referring to controls, but I think we're supposed to be thinking about the “constant” warming scenario. Please be careful about language usage here because control treatments were established earlier in the paper as being something else.”

Stability components were calculated as deviations from the control condition, as detailed in the Materials and Methods section, and in line with approaches used in other studies on ecological stability (Donohue et al. 2013 *Ecol. Lett.*; Hillebrand et al. 2018 *Ecol. Lett.*). For a given component of stability, comparisons among treatments were made indirectly, based on whether one treatment caused a significant deviation from the control while another did not. Therefore, the controls serve as the appropriate reference levels, rather than the fixed-pulses scenario, as they allow for the assessment of deviations from unperturbed conditions without any prior warming history.

“L249: missing word?”

The Reviewer is right. We added the “fixed-warming pulses” at the sentence now at line 214.

“L306: see comment above, I wouldn't describe this as “constant warming”.”

We agree with the Reviewer that the term "constant warming" may not accurately reflect the nature of the treatment. We have therefore changed the name of the treatment to "fixed-warming pulses" throughout the manuscript, as suggested by the Reviewer in the comment below.

“L335: Why was 70 minutes chosen? This seems a remarkably short exposure time to cause the kinds of treatment-level differences presented in the manuscript. I am quite surprised there were effects at all. After

how long were measurements then taken? Was it immediately after the 70-minute heating? If so, that could explain how treatment effects emerged despite the very short heat pulses.”

The duration of the treatment aligns with the duration of thermal anomalies observed at the study sites, and previous studies have shown that similar treatments are sufficient to cause a displacement in biofilm biomass compared to unmanipulated conditions (Fig. 1c). Measurements, as indicated at lines 354-355, were taken 5 to 7 days after the treatment application, allowing the biofilm adequate time to respond to experimental treatments (Dal Bello et al. 2015, *Glob. Change Biol.*). Although the treatment duration might seem too short to induce a response in terms of biomass and composition, it is important to note that the observed effects were not the result of a single temperature pulse, but rather the cumulative thermal history.

“L418: I find the explanation of the synchrony calculation to not be well enough explained. Please expand.”

Spatial synchrony was calculated for each plot as the average synchrony across MAGs, using Gross' metric, weighted by their relative abundance. We expanded the sentence to provide a clear explanation of how spatial synchrony was calculated (lines: 419-421).

“L431: I don't think this is really functional richness, but perhaps would be better defined as “functional group richness” since it is based on functional groupings. See the literature on functional groups for more info.”

We thank the Reviewer for the suggestion and have replaced the term “functional richness” with “functional group richness” where appropriate.

“Fig 1: I'm having a hard time understand the definition of constant warming used here. The authors seem not to be referring to a constant warming press perturbation, but rather pulse perturbations to the same temperature (have I understood correctly)? Calling this “constant warming” I think is a bit misleading as actually communities under a constant warming treatment are in this case no more frequently exposed to warmer temperatures than those under fluctuating temperatures. Perhaps a workaround would be to call these “fixed-warming heatwaves” or something similar, to make much clearer what these warming events actually represent.”

We changed the name from “constant warming” to “fixed-warming pulses” as suggested by the Reviewer.

“I also generally found figure 1 to be confusing, I did not follow the reasoning as written in the (wordy) blue boxes, and therefore was not convinced of why we would expect stability responses to as suggested. I wonder if replacing these with a more concrete example may help?”

Fixed-warming pulses are likely to favor taxa with similar thermal responses, leading to their dominance and the outcompetition of less adapted species, which subsequently reduces both taxonomic and functional diversity. When dominant taxa exhibit narrow thermal responses, synchronous fluctuations are expected, resulting in large biomass fluctuations and low temporal stability. The low functional redundancy in these communities further reduces their ability to buffer perturbations, making them highly sensitive to perturbations. In contrast, thermally fluctuating conditions can promote species coexistence through temporal niche differentiation, enhancing both functional and compositional diversity. Communities with greater diversity and functional redundancy are better equipped to buffer perturbations, with recovery driven by fast-growing taxa. However, intense environmental fluctuations may still favor stress-tolerant taxa, potentially reducing resilience due to trade-offs between stress tolerance and growth rates. While these more diversified

communities may exhibit greater stability over time and better buffer against extreme events, they often recover more slowly from disturbances. We have clarified this in Figure 1, adding further explanations in the insets and text (lines: 26-43 and 767-783).

“Fig 1a: “history”, “ambient temperature”, “reduce”, “resistance” are all misspelled.”

We corrected all the labels in Figure 1.

“Fig. 2: Were the control plots also exposed to extreme events? If not, the dip in panels a and b following the extreme events is interesting and perhaps worth discussing (some external environmental effect perhaps?). Also “temperature” is misspelled.”

As clarified in the Materials & Methods section and in previous responses, four unmanipulated plots received only the extreme temperature events, while the other four remained unmanipulated and served as controls. The small inflection observed for Hill’s 0 and 1 metrics in the controls after the extreme events appears to be natural variation rather than a result of external factors. This interpretation is further supported by the analysis of 16S-contigs, which revealed no changes following the extreme temperatures in the Hill 1 and 2 metrics (Supplementary Figure 4).

*“Fig. 2c,d: could the authors denote significant pairwise contrasts on the figure (e.g. using letters or * or similar)? This would improve the clarity of the figure.”*

We indicated significant pairwise contrasts in Figures 2, 3, 4, and 6 using asterisks to denote significance, accompanied by lines to highlight the comparisons.

“Fig. 3: “Temperature” is misspelled.”

We fixed the error in Figure 3.

“Fig. 5 and stability components: It may be helpful to readers unfamiliar with the specific stability components in the analysis if stability components were to all represent the same direction of interpretation. By which I mean e.g. high sensitivity = low stability, but high resilience = high stability. Could some of the axes be reversed or labels added to aid interpretation?”

We incorporated the Reviewer's suggestion by highlighting the directionality of the stability components. Arrows indicating whether the effect is stabilizing or destabilizing have been added to panels (c, d, f) of Figure 6.

Reviewer #2 (Remarks to the Author):

“In the manuscript entitled “Legacies of temperature fluctuations promote stability in marine biofilm communities”, the authors exposed microbial communities from biofilms in a rocky intertidal environment to different treatments of temperature to compare the effects of fluctuating vs. rather constant warming history on the response of the microbial communities toward extreme warming events. Overall, the study necessitated a lot of work, and presents very interesting results. The novel approach is well-designed, and the analyses are conducted carefully. I think that the results are of great significance to the field. My principal concern regards the structure of the manuscript. As the Results are described before the Material and Methods, some parts of the results feel more like M&M for me. I understand that these parts are necessary to understand the following results, but it makes the structure of the manuscript a little bit difficult to follow. As for now, the Results section is not only results, but presents also a lot of Materials and Methods, and even some part that are repetitions of the Introduction. I am not sure if it is possible to change the structure of the manuscript, but, in my opinion,

putting the Material and Method section before the Results and Discussion would greatly help gaining readability and understandability of the manuscript. Finally, I also have some specific comments that I listed hereafter.”

We appreciate the reviewer's recognition of the significance of our work and its relevance to a broad audience. We have adhered to the formatting guidelines of *Nature Communications*, which stipulate presenting the Results section before the Materials and Methods. This arrangement was chosen to enhance the logical flow and ease of interpretation of our study. In response to the reviewer's suggestions, we have streamlined the Materials and Methods section and removed non-essential details to improve the manuscript's clarity and conciseness. Further details on these revisions can be found in our responses to specific comments.

“L48. «of several ecosystems”. Maybe this part could be changed to “of many ecosystems”. Indeed, plenty of ecosystems were already shown to be affected by extreme climate events.”

DONE. We have replaced 'several' with 'many' as suggested by the Reviewer (line: 19).

“L88. “fluctuation warming conditions”. This should be change to “fluctuating warming conditions”. “

DONE. We have replaced fluctuation” with “fluctuating” as suggested by the Reviewer (line: 45).

“L128. “138 MAGs”. You should define MAG the first time the abbreviation is being used.”

DONE. We agree with the Reviewer and have defined MAGs as Metagenome Assembled Genomes at lines 88-89.

“L133. Maybe the name of the test being used here (Linear Mixed Effect Model) should be written in the parenthesis, at least the first time some results of such tests are described.”

While we recognize the importance of specifying the statistical analyses used in our study, we chose not to include these details in the manuscript to maintain readability. Additionally, the statistical results are reported in accordance with APA (American Psychological Association) style, which does not require naming the statistical tests used.

“L152-154. This is a repetition to what was already presented in the Introduction section.”

“L154-161. Similarly, this part does not belong to the Results section, but rather to M&M.”

DONE (response to both specific comments). We have removed these sections as suggested by the Reviewer to avoid repetition and to limit the inclusion of excessive methodological details in the Results section.

“L166-168. Again, this belongs to the Discussion section, not results.”

Although not strictly belonging to the Results section, we have decided to keep this sentence (lines: 125-128) as it reinforces the key takeaway in each section of the results, which we believe is important for clarity in complex studies with multiple analyses.

“L174-176. This is a repetition of the hypotheses presented in the Introduction section.”

DONE. We have removed this part as suggested by the Reviewer to avoid repetition and improve the manuscript's readability.

“L195-196. This does not belong to Results, rather to either Introduction or Discussion sections.”

DONE. We have removed this part as suggested by the Reviewer, as it was out of context and not strictly necessary in this section of the manuscript.

“L249. “our research revealed played a crucial role in augmenting biomass”. A word is missing, please reformulate.”

DONE. The reviewer is correct, and we have revised the sentence to include 'that fixed-warming pulses'. The updated sentence now reads: “Our research revealed that fixed-warming pulses played a crucial role in augmenting biomass” (line: 211).

“L254. “ High fluctuating warming” should be replaced by “Highly fluctuating warming”.”

DONE. We changed the sentence as suggested by the Reviewer (line: 217).

“L158-260. I agree, this history is far too often neglected in current research.”

We appreciate the Reviewer's recognition of the relevance of historical context in determining responses to future perturbations. This is a timely topic, particularly in understanding how ecological systems will respond to future challenges in the light of climate change, an area that is often underexplored in current research.

“L307. “On the other end” should be replaced by “On the other hand”. In addition, this must be used with “on the one hand” first.”

DONE. We replaced “On the other end” with “In contrast” that is more grammatically appropriate (line: 282).

“L308. Should “susceptibility” be replaced by “sensitivity”?”

DONE. We replaced “susceptibility” with “sensitivity” as suggested by the Reviewer (line: 284).

“L309-311. Keep in mind that you only studied microbial communities from a rocky intertidal area. While the results are very interesting, any extrapolation of the results should be done with caution, as microbial communities from different environment, and therefore with different community composition, may respond in another way to warming and unpredictability.”

The Reviewer raised an important point. We acknowledge that our conclusions and implications are specific to marine biofilm communities and should be cautiously extended to other microbial systems, as clarified at lines 285-288. Furthermore, we emphasize the need for further research to evaluate the broader applicability of our findings to other microbial systems (lines: 292-294).

“L335. Why 70 minutes? It is a very short duration, compared to what can occur in natural environments.”

The Reviewer is correct that 70 minutes might seem like a short duration. However, our in-situ air temperature data from the study site shows that, under natural conditions, these thermal events typically last around 1 to 2 hours, primarily during the central part of the day (the inset of Fig. 1c). Additionally, the distinction between these warming pulses and extreme temperatures was also based on the different durations of the thermal events (70 vs. 120 minutes). This approach reflects realistic environmental conditions and allows us to simulate the natural variability in thermal exposure experienced by the organisms at our study site.

“L336. What chambers? Please be more precise in the description of how the warming treatments were performed on the biofilms. Did the chambers let light go through during the warming treatments?”

We added a detailed description of the warming devices, which consisted of an aluminum box equipped with an adjustable butane stove and an opening to regulate heating (lines: 328-329). The chambers mostly prevented light from passing through, though this varied depending on the degree of the opening. To assess the potential effect of sunlight reduction during the warming sessions, we established control plots for artifacts, maintained in shaded conditions without heating for the entire duration of the warming sessions. This was achieved using cardboard chambers of the same size and shape as the aluminum heating chambers (lines: 336-337). No artifacts due to the warming chambers were detected for any of the response variables in our study (Supplementary Tables 8, 15 and 18).

“L326-343. How did you make sure that all plots (the warmed and the control ones) were similar in composition at the beginning of the experiment?”

Experimental plots were randomly allocated to different experimental conditions and ensure that treatments plots were interspersed across the study area (lines: 312-316). This procedure should ensure the plots were similar in composition at beginning of the experiment. Furthermore, the observation that most response variables, such as richness and Shannon diversity, exhibited similar values at the first sampling date — after the application of the first warming pulse — indicates that the plots were indeed similar in composition at the beginning of the study.

“L360. Why are the references not in the same format as the others?”

DONE. We standardized the reference style of both references according to the journal's guidelines (line: 342).

“L415-420. I believe these analyses were performed with R?”

DONE. These analyses were all performed using R. We indicated the specific R packages utilized at lines 417 and 423.

“L434. “Muillot et al.” should be replaced by “Mouillot et al.”.”

DONE. We have corrected the misspelling (line: 437).

“L725 and L734. I would replace “depauperated” by “less diverse”.”

DONE. We have replaced the “depauperated” with “less diverse” (lines: 770 and 779)

“L736. “indicate” should be “indicates”.”

DONE. We changed the word "indicate" to the third person (line: 781).

“L753, L760, L770. Please refer here to the fluctuating sequences as “fluct-s1”, “fluct-s2”, and “fluct-s3”, to be consistent with other parts of the manuscript.”

DONE. We changed the names of fluctuating sequences as suggested by the Reviewer and ensured they were consistent throughout the manuscript.

“L771. “and is expressed as community-level weighted mean”. The “as” is missing.”

DONE. We corrected the error and added the word “as” to the sentence (line: 836).

Reviewer #3 (Remarks to the Author):

“The ms by Rindi et al. uses microbial communities that form biofilms on intertidal rocky shores to experimentally examine effects of constant and fluctuating temperature stress on microbial community diversity and function, and how their responses to these different stressors affect overall responses to an extreme temperature event, with particular focus on stability (resistance, resilience and temporal stability) of biofilm biomass.

This work is timely as although there has been significant research on the effect of different types of temperature stress (or other constant vs variable stressors) on ecological communities, how these effects then affect subsequent responses to stressors has been largely overlooked. This is important because such information is needed in order to better predict the fate of biodiversity and ecosystem function under future environmental change.

Importantly, this work quantitatively examines key aspects of stability of communities: resistance to stress, resilience and temporal stability, and uses information of diversity (who are they?) and function (what can they do?) of multiple members in the microbial community to test hypotheses about functional redundancy and other mechanisms which may explain overall stability in the context of stress.

Besides the rationale, importance and ease of manipulation of the study system and the environmental context (e.g. temperatures based on realistic predictions), there are two key aspects that make this experiment very relevant: (i) it is done in the field, in the context of natural communities and natural variation in factors other than those manipulated here; (ii) the experimental design is appropriate and rigorous (e.g. replication of fluctuating treatments, procedural controls), enabling rigorous tests of the proposed hypotheses.”

We express our thankfulness to the Reviewer for her/his appreciation of our work and for acknowledging the significance of testing ecological theory through rigorous field-testing based on a solid experimental design.

“The main issue with this work is the data used to characterise microbial community diversity and (potential) function. The authors used metagenomic sequencing to then assemble metagenome-assembled genomes (MAGs), which resulted in 138 MAGs, or microbial “species” for which they also have information on their potential functional role – ‘potential’ because one does not know whether the genes observed in their genome are being expressed or not, as no RNA/gene expression was quantified in this study. A key question is whether those 138 MAGs are representative of the diversity and functional potential of the entire microbial community. These communities typically have many hundreds to thousands of taxa (ASVs characterised by e.g. 16S rRNA gene amplicon sequencing) and functional genes (via metagenomics). While MAGs allows understanding which functional genes are present in which taxa, and thus better tackle the functional redundancy issue, this study should have also used information of taxonomic and functional gene diversity to test hypotheses about

changes in diversity and function in response to treatments. For example, would the patterns observed in richness or other diversity measures remain if data of the entire community (eg via 16S sequencing) were used?"

We appreciate the Reviewer's concern regarding the absence of a 16S rRNA metagenomic sequencing approach for characterizing compositional changes, which indeed provides a more detailed representation of taxonomic diversity, particularly among rare taxa. While we did not initially include this analysis, we have since assembled all 16S rRNA reads from our filtered and trimmed data to derive 16S contigs (lines: 381-390). These contigs serve as proxies for taxa, helping to overcome the challenge of having 16S rRNA reads from different regions of the 16S rRNA gene. To quantify each 16S contig, we mapped back the 16S reads using the assembled 16S contigs as references. The analysis produced 2,097 distinct contigs, consistent with findings on OTU richness from a previous study at the same site using 16S rRNA amplicon sequencing (Maggi et al. 2017, *Mar. Ecol. Prog. Ser.*). Although there are some differences, the Hill numbers calculated from the 16S contigs display a pattern consistent with that observed from MAGs. Notably, plots subjected to a fluctuating warming regime showed divergent trends compared to control and fixed warming treatment, suggesting that fluctuating temperatures selectively filtered out thermally sensitive taxa while favoring more resistant ones. We have included these findings in the Results section (lines: 102-109 and Supplementary Figure 4).

While this analysis is not entirely equivalent to classical 16S amplicon sequencing, our primary objective was to characterize the compositional changes that underlie the stability of the biofilm community, which is often driven by the most abundant taxa. Given the capability of whole-genome sequencing to accurately capture the diversity of potential functions within the most representative taxa, we believe this approach is well-suited to achieving our study's goals.

"A second important issue has to do with the ecological trait assignment to MAGs/genes. How accurate is this for this system? These assignments are based on databases which tend to be system-specific and a lot of understanding of particular taxa, their functions and roles in that system are needed to accurately predict/assign ecological traits. If the ecological traits assigned are not accurate, then this will have implications for the observed functional responses and the interpretation of the data. This is more problematic where actual microbial function (eg via gene expression or proteomics/metabolomics) has not been measured to verify whether assigned traits make sense."

The Reviewer raised an important concern regarding our methodology's focus on the potential functions of different MAGs. We have clarified this in the manuscript, emphasizing that our approach is not only identifying potential functions but is also robust in doing so. Our methodology relies on the assignment of specific gene pathways tied to well-characterized functions (e.g., stress tolerance, energy acquisition) as documented extensively in the literature. These gene pathways are rigorously cross-referenced with the KEGG database (with accuracy larger than 99.99%), ensuring that only well-established and validated pathways are included. This approach is deliberately restrictive, focusing on accurate gene-function relationships, and is designed to minimize the risk of misassigning functions. We also included in Methods a note of caution in the interpretation of the results deriving from the functional characterization (lines: 405-411). We acknowledge the inherent limitations of any functional prediction method, but we believe that our approach, especially when combined with our carefully controlled experimental setting, provides a reliable characterization of the functional potential of the MAGs. This methodology offers a high level of confidence in the functions identified, allowing for meaningful insights into the ecological roles of biofilm community.

"I think the authors need to address the two major points I raise above in the Discussion. Perhaps also the structure of the ms could be modified so that biofilm stability is presented first and then analyses of microbial diversity and function come after as potential mechanisms that may explain effects on biofilm stability. I also suggest revising the Introduction as some parts are a bit repetitive."

We have chosen to maintain the original structure of the manuscript, as it follows the logical progression of our hypothesis. The warming history alters the compositional and functional structure of the biofilm, which subsequently drives changes in multiple components of stability. However, if the Editor and Reviewer believe that restructuring could improve the readability of the manuscript, we are open to considering and discussing this possibility. Additionally, we have incorporated the Reviewer's suggestion to make the Introduction less repetitive and have sharpened the focus on the role of historical conditions in shaping the response of ecological communities to future perturbations.

Minor comments

“Lines 67-70: Sentence too long. Suggest ending after ‘natural communities’ and deleting the rest as its not clear.”

As we simplified the Introduction this sentence is no longer part of the current version of the manuscript.

“L 78-81: It is worth mentioning here that there could also be trade-offs among stressors, i.e. that traits that lead to resistance to one stressor (stressor ‘A’) may lead to lower resistance to another stressor (‘B’). In the context of multiple-stressors this is a very important point which may impact diversity/stability relationships.”

We agree with the Reviewer that this is an important aspect to consider, as ecological communities face multiple stressors, and gaining resistance to one might increase sensitivity to another, potentially impacting different dimensions of stability. We have taken the Reviewer's suggestion on board; however, instead of addressing it in the Introduction, we have included this aspect in the Discussion section (lines: 265-268).

“L 83-86: repetitive from above (eg 71-73).”

We streamlined the introduction and revised the closing sentence of the paragraph to avoid repetition with the opening sentence (lines: 26-28 and lines: 41-43).

“L 88: “fluctuating””

We have corrected the error as indicated by the Reviewer.

“L 120: include SD, state these were multiple fluctuating treatments and refer to Methods for the rationale”

We included the standard deviation (SD) value for the fluctuating scenario and clarified that the fluctuating scenario consists of three distinct fluctuating treatments. We have also referred to the Methods section for the rationale behind this approach (lines: 76-78).

L 128: were there 138 or 137 MAGs as per lines L 392, 397?

The total number of MAGs was 137. We corrected the error on line 88.

“L151: effect “of” fluctuating temperatures on...”

We have corrected the error and replaced “on” with “of” (line: 87).

In our document, the reviewers' comments appear in italics, while our responses are in normal text. Additionally, any amendments made in the main document in response to the reviewers' comments are highlighted in red.

Reviewer #1

"I previously reviewed this manuscript by Rindi et al. as reviewer 1 and find this version much improved. I appreciated their detailed and thorough response to reviewer comments, and now have only minor comments below. I do think the figures could be further modified to clarify that the extreme-event-only treatment didn't differ from control before manipulation (as shown in the response letter). Whether in the main manuscript or at least in the supplement, this further addition would be useful to many readers I think."

We thank the Reviewer for their appreciation of our work in revision process. Following their suggestion, we have added time series of diversity and functional metrics for controls and extreme-only treatments over the entire duration of the study as Supplementary Figures 5, 6, and 8 in the Supplementary Information.

Specific comments:

"L63: what does "diversified" mean here? I think the authors might be referring to the concept of response diversity – a wider variety of different responses to environmental change – but am not sure. Or do you simply mean biodiversity (species richness)?"

We were referring to communities characterized by a high number of species. To clarify this point, we revised the sentence to "In species-rich communities" (line: 29).

"L138-141: If I understand this correctly, this addresses my previous comment about only considering the extreme event treatments after the events. In your response to reviewers, you offered to include the figures in the manuscript. Indeed I think that would be useful here (in the supplement) and perhaps also mapped onto Fig. 1c."

As suggested by the Reviewer, we have included plots of taxonomic and functional diversity, showing separate trajectories for controls and the extreme-only condition, in the Supplementary Information (Supplementary Figures 5, 7, and 8).

"L178-179: sorry if I missed it but what is "compositional diversity"? Do you mean compositional dissimilarity (which would be beta diversity). As far as I've always understood the concept, composition doesn't have diversity – unless we're talking about diversity in space (between patches)?"

DONE. We were referring to taxonomic diversity and initially used "compositional," a less common but broader term that accounts for species combinations and their relative frequencies within a community. However, we agree with the Reviewer that this term may be ambiguous or overly technical. To enhance clarity, we have replaced it with "taxonomic diversity" (lines 145–146).

"L212: maybe worth citing the Hillebrand et al paper here to clarify what you mean."

DONE. We have added the reference to Hillebrand et al. (2018, Ecology Letters) as indicated by the Reviewer (line: 179).

"L240: "taxonomic" I think"

DONE. We have changed "taxonomical" to "taxonomic" as suggested (line: 207).

"L314: "rather than reducing stability" seems like a separate idea here. Is the opposite of constant thermal warming a reduction in stability? Which dimension of stability are we talking about? Consider rephrasing this sentence"

Thank you for your comment. We have revised the sentence to clarify that the outcome observed contrasts with our initial hypothesis and prevailing expectation. The new wording states that, contrary to the prevailing expectation, a relatively constant warming regime led to reduced sensitivity (lines: 278-281). We believe this change directly addresses your concern and enhances the clarity of our argument.

“L315: I don’t know if I agree that warming can “provide a broader niche space”. Warming is by definition just shifting the position along an environmental axis, and so will benefit species with her thermal optima, while disadvantaging others. Please consider rephrasing this whole sentence or clarifying these ideas.”

Thank you for your comment. I understand your point. However, under variable conditions—albeit not extreme—different species with distinct thermal optima may be alternately favored over time, thereby expanding the community's niche space and promoting species coexistence. I have rephrased the sentence (lines 278–281) to clarify that even relatively constant conditions may generate subtle environmental variations over time, ultimately facilitating coexistence among taxa.

“I noticed that Dal Bello et al is cited as two different references. There may be other similar issues, but I did not check in detail.”

DONE. We have revised and removed any duplicated references.

Reviewer #2

“I want to thank the authors for having taken the time to carefully answers all my comments on the previous version of the manuscript. I think that the manuscript is clearer now, notably why and how the experimental manipulations were implemented is now much clearer.”

We thank the Reviewer for her/his suggestion, which has improved the readability of the manuscript.

Reviewer #3

“The authors have done a good job at addressing my (and other Reviewer’s) comments. I think the inclusion of analyses of the 16S rRNA gene contigs provides more robustness on the interpretation of the data. My main comment is that there should be 1 sentence in the Discussion that states, up front, that when referring to a community in your inferences – particularly functon(as the inclusion of the 16S data now captures most of the taxonomic composition) – you are focusing on dominant taxa obtained via MAGs.”

Thank you for your comment. We appreciate that you recognized the value of our work and our approach to better characterize the community's diversity. We agree that our analysis focuses on the dominant taxa, and we will specify this clearly throughout the manuscript.

Some minor comments below:

“Reviewer 1 made an interesting comment with regards to presenting data from the extreme plots before the application of the treatment. I agree with the response from the authors but I think it would be useful to include these (in Supplementary?) data given that the authors have it and have already analysed it in their response to the Reviewer’s comment.”

We have incorporated the reviewers’ suggestions by including the time series of diversity and functional metrics for controls and extreme-only treatments over the entire study period in the Supplementary Information (see Supplementary Figures 5, 7, and 8).

“Line 101: to ‘the’ biofilm community.”

DONE. We have added "the" to the sentence (line 68).

"154 and throughout: check that the terminology used now – 'fixed' – has been consistently changed as some contrasts and statistical tables still refer to 'const'."

DONE. We thank the Reviewer for their suggestion and have updated the terminology in both the manuscript and the Supplementary Information.

"L 160: rare what? taxa?"

DONE. We have added "taxa" at the end of the sentence.

"L 241: here and throughout the Discussion, rather than inferring patters for the entire community, specify that the focus was on some, likely key members of the community. E.g. "...functional structure of dominant members of a community."

DONE. We appreciate the Reviewer's comment on specifying the reference, particularly for functional metrics, which describe the dominant part of the community. We have implemented this suggestion at line 209 and throughout the discussion.

"L 416: join these two sentences."

DONE. We have merged the two sentences as suggested by the Reviewer (lines: 393–396).

"L 807: add 'inter-specific' before competition."

DONE. We have added "interspecific" before "competition" as suggested by the Reviewer (line: 765).

"Supplementary Fig. 6 b: x-axis should be 'fold' not 'old'."

DONE. We have updated the x-axis label of Supplementary Figure 6b in accordance with the Reviewer's comment.

"Sup. Table S1 (and others): None of the terms in the model are significant except for the intercept – why do you report post-hoc tests between treatments at the last sampling time? This would be appropriate if there was a significant effect of treatment in an analysis of data only from that time point or if there was an interaction between treatment x time (where the other time points would also need to be included in these contrasts). However, you would need to run an analysis of variance testing overall effects of terms in the model first, rather than each term against the intercept as you have done here, e.g. using the Anova function in the R car package on this GLMM model (e.g. by fitting the model using the lmer function in the lme R package). I suggest you do this for all similar analyses."

We appreciate the Reviewer's concern regarding our model structure. Our contrasts at the final sampling date were intentionally designed to assess the effect of warming history relative to the control at the end of the experiment, which is central to our hypothesis. Similarly, the analysis centered on the second sampling date was conducted to evaluate the immediate effects of warming history following the application of the warming regimes – as clearly evidenced by reporting the outcome as a z-test on the intercept rather than using ANOVA. In contrast, the tests at the final sampling date specifically examined how extreme events influenced the recovery of biofilm community. We clarified this point in Methods at lines: 470-472.

The absence of a significant treatment \times time interaction for some variables suggests that deviations from the control were not sufficiently strong or consistent across time points to yield significant interactions. However, we acknowledge that the limited number of time points may have restricted

our ability to capture some interaction effects, leading to detectable deviations only at the final sampling date.

Additionally, to prevent any misinterpretation, we have modified the title in the Supplementary Table of contrasts by removing the term 'post-hoc.' Our contrasts were defined a priori based on our hypotheses, not in a post-hoc manner (lines: 466-470)

We believe that our analytical framework accurately reflects a hypothesis-driven approach. While alternative methods such as ANOVA could be considered, our current model structure is best suited to directly test the effects of warming history at key time points and assess the impact of extreme events on biofilm community dynamics.